# Fragment-based Pretraining and Finetuning on Molecular Graphs

**Kha-Dinh Luong, Ambuj Singh**
Department of Computer Science
University of California, Santa Barbara
Santa Barbara, CA 93106
{vluong,ambuj}@cs.ucsb.edu

## Abstract

Property prediction on molecular graphs is an important application of Graph Neural Networks (GNNs). Recently, unlabeled molecular data has become abundant, which facilitates the rapid development of self-supervised learning for GNNs in the chemical domain. In this work, we propose pretraining GNNs at the fragment level, a promising middle ground to overcome the limitations of node-level and graph-level pretraining. Borrowing techniques from recent work on principal subgraph mining, we obtain a compact vocabulary of prevalent fragments from a large pretraining dataset. From the extracted vocabulary, we introduce several fragment-based contrastive and predictive pretraining tasks. The contrastive learning task jointly pretrains two different GNNs: one on molecular graphs and the other on fragment graphs, which represents higher-order connectivity within molecules. By enforcing consistency between the fragment embedding and the aggregated embedding of the corresponding atoms from the molecular graphs, we ensure that the embeddings capture structural information at multiple resolutions. The structural information of fragment graphs is further exploited to extract auxiliary labels for graph-level predictive pretraining. We employ both the pretrained molecular-based and fragment-based GNNs for downstream prediction, thus utilizing the fragment information during finetuning. Our graph fragment-based pretraining (GraphFP) advances the performances on $5$ out of $8$ common molecular benchmarks and improves the performances on long-range biological benchmarks by at least $11.5\%$. Code is available at: https://github.com/lvkd84/GraphFP.

## 1 Introduction

The rise of Graph Neural Networks (GNNs) has captured the interest of the computational chemistry community [11]. Representing molecules and chemical structures as graphs is beneficial because the topology and connectivity are preserved and can be directly analyzed during learning [13], leading to state-of-the-art performance. However, there is a problem in that while highly expressive GNNs are data-hungry, the majority of molecular datasets available are considerably small, posing a critical challenge to the generalization of GNNs. Following the steps that were taken in other domains such as text and images, a potential solution is to pretrain GNNs on large-scale unlabeled data via self-supervised pretraining [3, 5]. How to do so effectively is still an open question since graphs are more complex than images or text, and the chemical specificity of molecular graphs needs to be considered when designing graph-based pretraining tasks.

A large number of pretraining methods have been proposed for molecular graphs [38, 40]. For example, [15] uses node embeddings to predict attributes of masked nodes, [17, 27] predicts graph structural properties, and [16] generatively reconstructs graphs via node and edge predictions. Another

37th Conference on Neural Information Processing Systems (NeurIPS 2023).

popular direction is contrastive learning [26, 32, 35, 45, 46], in which multiple views of the same graph are mapped closer to each other in the embedding space. These views are commonly generated via augmentations to the original graphs [35, 45, 46]. However, without careful adjustments, there is a great risk of violating the chemical validity or changing the chemical properties of the original molecular graph after applying the augmentations. Other methods such as [23, 31] avoid this problem by contrasting 2D-graphs against 3D-graphs. Unfortunately, privileged information such as 3D coordinates is often expensive to obtain. Node versus graph contrastive learning has also been investigated [32], however, this approach may encourage over-smoothing among node embeddings.

In general, the majority of works pretrain embeddings at either node-level or graph-level. Node-level pretraining may be limited to capturing local patterns, neglecting the higher-order structural arrangements while graph-level methods may overlook the finer details. As such, motif-based or fragment-based pretraining is a new direction that potentially overcomes these problems [27, 48, 49]. Still, existing fragment-based methods use either suboptimal fragmentation or fragmentation embeddings. GROVER [27] predicts fragments from node and graph embeddings, however, their fragments are k-hop subgraphs that cannot account for chemically meaningful subgraphs with varying sizes and structures. MICRO-Graph [48] contrastively learns subgraph embeddings versus graph embeddings; however, these embeddings may not effectively capture global patterns. MGSSL [49] utilizes graph topology via depth-first search or breadth-first search to guide the fragment-based generation of molecules; however, their multi-step fragmentation may overly decompose molecules, thereby losing the ability to represent higher-order structural patterns.

In this paper, we propose GraphFP, a novel fragment-level contrastive pretraining framework that captures both granular patterns and higher-order connectivity. For each molecule, we obtain two representations, a molecular graph and a fragment graph, each of which is processed by a separate GNN. The molecular GNN learns node embeddings that capture local patterns while the fragment GNN learns fragment embeddings that encode global connectivity. The GNNs are jointly trained via a contrastive task that enforces consistency between the fragment embedding and the aggregated embedding of the corresponding atoms from the molecular graphs. Unlike previous work, we pretrain the fragment-based aggregation of nodes instead of individual nodes. With the fragment embeddings encoding the global patterns, contrasting them with the aggregation of node embeddings reduces the risk of over-smoothing and allows flexibility in learning the appropriate node latent space. Since aggregation of nodes in this context can also be considered a kind of fragment representation, our framework essentially contrasts views of fragments, each of which captures structural information at a different scale. Moreover, as the molecular graphs and the fragment graphs are chemically faithful, our framework requires no privileged information or augmentation.

Exploiting the prepared fragment graphs, we further introduce two predictive pretraining tasks. Given a molecular graph as input, the molecular GNN is trained to predict structure-related labels extracted from the corresponding fragment graph. These tasks enhance the structural understanding of the molecular GNN and can be conducted together with contrastive pretraining.

To generate a vocabulary of molecular fragments, we utilize Principal Subgraph Mining [20] to extract optimized prevalent fragments that span the pretraining dataset. Compared to fragmentations used in previous works, this approach can produce a concise and diverse vocabulary of fragments, each of which is sufficiently frequent without sacrificing fragment size.

We evaluate GraphFP on benchmark chemical datasets and long-range biological datasets. Interestingly, both the molecular and the fragment GNNs can be used in downstream tasks, providing enriched signals for prediction. Empirical results and analysis show the effectiveness of our proposed methods. In particular, our pretraining strategies obtain the best results on 5 out of 8 common chemical benchmarks and improve performances by 11.5% and 14% on long-range biological datasets.

## 2 Related Works

### 2.1 Representation Learning on Molecules

Representation learning on molecules has made use of fixed hand-crafted representations such as descriptors and fingerprints [30, 43], string-based representations such as SMILES and InChI [22, 28], and molecular images [47]. Currently, state-of-the-art methods rely on representing molecules as graphs and couple them with modern graph-based learning algorithms like GNNs [11]. Depending

on the type of compounds (small molecules, proteins, crystals), different graph representations can be constructed [11, 39, 44]. Compared to graph-based representation, molecular descriptors and fingerprints cannot encode structural information effectively while string-based representations require an additional layer of syntax which may significantly complicate the learning problem.

## 2.2  Graph Neural Networks

Given a graph $G = (V, E)$ with node attributes $x_v$ for $v \in V$ and edge attributes $e_{uv}$ for $(u, v) \in E$, GNNs learn graph embedding $h_G$ and node embedding $h_v$. At each iteration, a node updates its embedding by gathering information from its neighborhood, including both the neighboring nodes and the associated edges. This process often involves an aggregating function $M_k$ and an updating function $U_k$ [11]. After the $k$-th iteration (layer), the embedding of node $v$ is, $h_v^{(k)} = U_k(h_v^{(k-1)}, M_k(\{(h_v^{(k-1)}, h_u^{(k-1)}, e_{uv}) | u \in N(v)\}))$, where $N(v)$ is the set of neighbors of $v$ and $h_v^{(0)} = x_v$. The aggregating function $M_k$ pools the information from $v$'s neighbors into an aggregated message. Next, the updating function $U_k$ updates the embedding of $v$ based on its previous embedding $h_v^{(k-1)}$ and the aggregated message. An additional readout function $R$ combines the final node embeddings into a graph embedding $h_G = R(\{h_v^{(K)} | v \in V\})$, where $K$ is the number of iterations (layers). Since there is usually no ordering among the nodes in a graph, the readout function $R$ is often chosen to be order-invariant [41]. Following this framework, a variety of GNNs has been proposed [12, 34, 41], some specifically for molecular graphs [9, 29].

## 2.3  Pretraining on Graph Neural Networks

To alleviate the generalization problem of graph-based learning in the chemical domain, graph pretraining has been actively explored [15, 23, 26, 31, 32, 35, 45, 46] in order to take advantage of large databases of unlabeled molecules [10].

In terms of the pretraining paradigm, existing methods can be categorized into being predictive [15, 27, 37], contrastive [23, 26, 31, 35, 45, 46], or generative [16, 49]. Predictive methods often require moderate to large labeled datasets for pretraining. In the case of an unlabeled large dataset, chemically or topologically generic labels can be generated. Despite the simple setup, predictive methods are prone to negative transfer [15]. On the other hand, contrastive methods aim to learn a robust embedding space by using diverse molecular views [23, 31, 35, 45, 46]. Generative methods intend to learn the distribution of the components constituting molecular graphs [16, 49].

At the pretraining level, methods can be sorted into node-level [16, 27], graph-level [15, 23, 26, 31, 35, 45, 46], and more recently motif-level [27, 48, 49]. Node-level methods only learn chemical semantic patterns at the lowest granularity, limiting their ability to capture higher-order molecular arrangements, while graph-level methods may miss the granular details. Motif-based or fragment-based pretraining has emerged as a possible solution to these problems [27, 48, 49]; however, existing methods use suboptimal fragmentation or suboptimal fragment embeddings. In this work, we learn fragment embeddings that effectively capture both local and global topology. We exploit the fragment information at every step of the framework, from pretraining to finetuning.

## 3  Methods

In this section, we introduce the technical details of our pretraining framework GraphFP. We begin with discussing molecular fragmentation and fragment graph construction based on an extracted vocabulary. Then, we describe our fragment-based pretraining strategies, including one contrastive task and two predictive tasks. Finally, we explain our approach for combining pretraining strategies and the pretrained molecular-based and fragment-based GNNs for downstream predictions.

### 3.1  Molecule Fragmentation

Graph fragmentation plays a fundamental role in the quality of the learning models because it dictates the global connectivity patterns. Existing methods rely on variations of rule-based procedures such as BRICS [4] or RECAP [21]. Though chemistry-inspired, the extracted vocabulary is often large, to the order of the size of the pretraining dataset, and contains unique or low-frequency fragments,

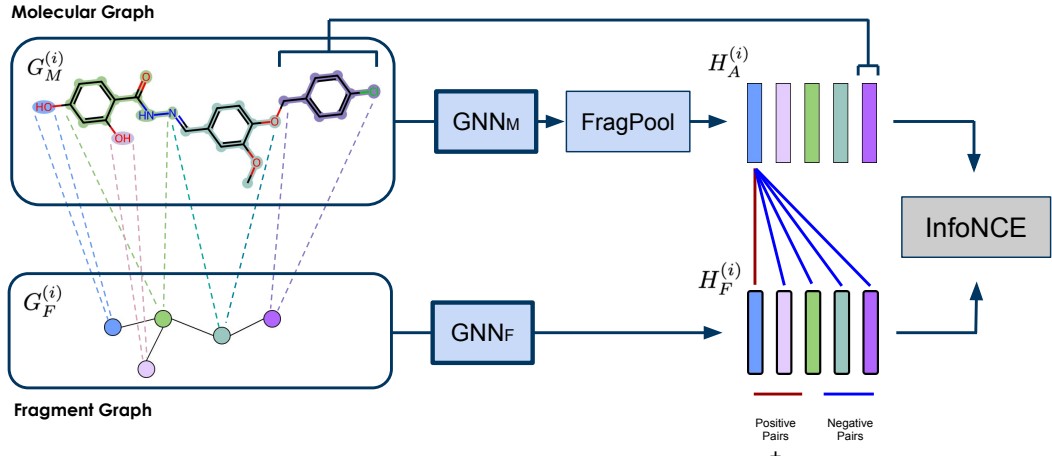

Figure 1: Fragment-based contrastive pretraining framework. $\text{GNN}_M$ processes molecular graphs while $\text{GNN}_F$ processes fragment graphs. The fragment-based pooling function FRAGPOOL aggregates node embeddings into a combined embedding that forms a positive contrastive pair with the corresponding fragment embedding. Notice that the two -OH groups, blue and light pink, are considered distinct. Therefore, the aggregated node embedding corresponding to the blue fragment and the embedding of the light pink fragment form a negative pair.

posing a significant challenge for pattern recognition. To overcome these drawbacks, some works further break down fragments [49], even to the level of rings and bonds [18]. However, reducing the sizes of fragments forfeits their ability to capture higher-order arrangements.

### 3.1.1 Principal Subgraph Extraction

In this work, we borrow the Principal Subgraph Mining algorithm from [20] to alleviate the above-mentioned problems. Given a graph $G = (V, E)$, a subgraph of $G$ is defined as $S = (\tilde{V}, \tilde{E})$, where $\tilde{V} \subseteq V$ and $\tilde{E} \subseteq E$. In a vocabulary of subgraphs, $S$ is a principal subgraph if $\forall S'$ intersecting with $S$ in any molecule, either $S' \subseteq S$ or $c(S') \leq c(S)$ where $c(\cdot)$ counts the occurrences of a fragment among molecules. Intuitively, principal subgraphs are fragments with both larger sizes and more frequent occurrences. The algorithm heuristically constructs a vocabulary of such principal subgraphs via a few steps:

**Initialize:** Initialize the vocabulary with unique atoms.

**Merge:** For each molecular graph, for all pairs of overlapping fragments in the graph, merge the fragment in the pair and update the occurrences of the resulting combined fragment.

**Update:** Update the vocabulary with the merged fragment with the highest occurrence. Repeat the last two steps until reaching the predefined vocabulary size.

For a more detailed description, we refer readers to [20]. We investigate the effect of vocabulary sizes on the performance of pretrained models in Section 4.4 and Appendix C. Similarly, we investigate the effect of fragmentation strategies in Section 4.2.

### 3.1.2 Fragment-graph Construction

Given an extracted vocabulary of principal fragments, for each molecular graph, we construct a corresponding fragment graph. Let $F = \{S^{(0)}, S^{(1)}, ..., S^{(m)}\}$ be the fragmentation of a molecular graph $G_M = (V_M, E_M)$, where $S^{(i)} = (\tilde{V}^{(i)}, \tilde{E}^{(i)})$ is a fragment subgraph, $\tilde{V}^{(i)} \cap \tilde{V}^{(j)} = \emptyset$, and $\cup_{i=1}^{m} \tilde{V}^{(i)} = V_M$. We denote the fragment graph as $G_F = (V_F, E_F)$, where $|V_F| = |F|$ and each node $v_F^{(i)} \in V_F$ corresponds to a fragment $S^{(i)}$. An edge exists between two fragment nodes of $G_F$ if there exists at least a bond interconnecting atoms from the fragments. Formally,

$E_F = \{(i,j)|\exists u, v, u \in \tilde{V}^{(i)}, v \in \tilde{V}^{(j)}, (u,v) \in E\}$. For simplicity, in this work, we withhold edge features in the fragment graph. As a result, a fragment graph purely represents the higher-order connectivity between the large components within a molecule. Fragment node features are embeddings from an optimizable lookup table. Vocabulary and fragment graphs statistics are provided in Appendix C.

## 3.2 Fragment-based Contrastive Pretraining

Figure 1 illustrates our contrastive framework. We define two separate encoders, $\text{GNN}_M$ and $\text{GNN}_F$. $\text{GNN}_M$ processes molecular graphs and produces node embeddings while $\text{GNN}_F$ processes fragment graphs and produces fragment embeddings. Since GNNs can capture structural information, the node embeddings encode local connectivity of neighborhoods surrounding atoms. Similarly, the fragment embeddings encode global connectivity and positions of fragments within the global context.

We apply contrastive pretraining at the fragment level. Contrastive pretraining learns robust latent spaces by mapping similar views closer to each other in the embedding space while separating dissimilar views. Learning instances are processed in pairs, in which similar views form the positive pairs while dissimilar views form the negative pairs. In this case, a positive pair consists of a fragment from a fragment graph and the collection of atoms constituting this fragment from the corresponding molecular graph. On the other hand, a fragment and any collection of atom nodes constituting a different fragment instance form a negative pair. Notice that different occurrences of the same type of fragment in the same or different molecules are considered distinct instances because when structural contexts are taken into account, the embeddings of these instances are dissimilar. The -OH groups in Figure 1 illustrate one such case.

To obtain the collective embedding of atom nodes corresponding to a fragment, we define a function $\text{FRAGPOOL}(\cdot)$ that combines node embeddings. Contrasting the combined embedding of nodes against the fragment embedding allows flexibility within the latent representations of individual nodes. Intuitively, the learning task enforces consistency between fragment embeddings and collective node embeddings, incorporating higher-order connectivity information. Through the learning process, the information is optimally distributed among the nodes so that each node only holds a part of this collective knowledge. Arguably, such setup is semantically reasonable because nodes and fragments represent different orders of connectivity. A node does not necessarily hold the same structural information as the corresponding fragment. Instead, it is sufficient for a group of nodes to collectively represent such higher-order information. Essentially, the fragment embeddings and the collective embeddings of atom nodes can be considered different views of molecular fragments.

Extending the notations in 3.1.2, given a pretraining dataset of molecular graphs $\mathcal{D}_M = \{G_M^{(1)}, G_M^{(2)}, ..., G_M^{(N)}\}$, we obtain a set of fragmentations $\mathcal{F} = \{F^{(1)}, F^{(2)}, ..., F^{(N)}\}$ and a set of corresponding fragment graphs $\mathcal{D}_F = \{G_F^{(1)}, G_F^{(2)}, ..., G_F^{(N)}\}$, with $G_M^{(i)} = (V_M^{(i)}, E_M^{(i)})$ and $G_F^{(i)} = (V_F^{(i)}, E_F^{(i)})$. More specifically, $V_M^{(i)}$ is the set of atom nodes in the molecular graph $G_M^{(i)}$ and $V_F^{(i)}$ is the set of fragment nodes in the fragment graph $G_F^{(i)}$. For the $i$-th molecule, let $H_M^{(i)} \in \mathbb{R}^{|V_M^{(i)}| \times d}$ and $H_F^{(i)} \in \mathbb{R}^{|V_F^{(i)}| \times d}$ be the final node embeddings and fragment embeddings after applying $\text{GNN}_M$ and $\text{GNN}_F$, respectively, i.e:

$$H_M^{(i)} = \text{GNN}_M(V_M^{(i)}, E_M^{(i)}) \tag{1}$$

$$H_F^{(i)} = \text{GNN}_F(V_F^{(i)}, E_F^{(i)}) \tag{2}$$

We further compute the fragment-based aggregation of node embeddings:

$$H_A^{(i)} = \text{FRAGPOOL}(H_M^{(i)}, F^{(i)}), \tag{3}$$

where $H_A^{(i)} \in \mathbb{R}^{|V_F^{(i)}| \times d}$ has the same dimensions as those of $H_F^{(i)}$. The $r$-th rows, $h_{A,r}^{(i)} \in H_A^{(i)}$ and $h_{F,r}^{(i)} \in H_F^{(i)}$, from both matrices are embeddings of the same fragment from the original molecule. In our contrastive framework, $h_{F,r}^{(i)}$ is the positive example for the anchor $h_{A,r}^{(i)}$. The negative learning examples are sampled from:

$$\mathcal{X}_{i,r}^- = \{H_{F,q}^{(j)})|j \neq i \vee q \neq r\} \tag{4}$$

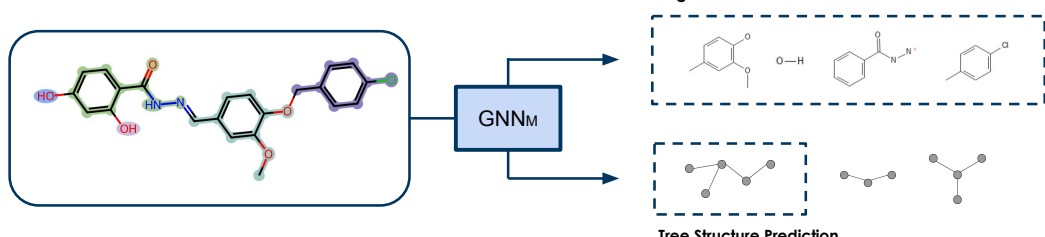

Figure 2: Fragment-based predictive pretraining. $\text{GNN}_M$ processes molecular graphs and produces graph-level embeddings used for prediction. The right upper box shows unique fragments that exist in the input molecule while the right lower box shows the ground-truth structural backbone. A few other backbones are also visualized for comparison.

We minimize the contrastive learning objective based on the InfoNCE loss [25]:

$$\mathcal{L}_C = -\mathbb{E}_{i,r}\left[\frac{\exp(\langle h_{A,r}^{(i)}, h_{F,r}^{(i)}\rangle)}{\exp(\langle h_{A,r}^{(i)}, h_{F,r}^{(i)}\rangle) + \sum_{h^- \sim \mathcal{X}_{i,r}^-}\exp(\langle h_{A,r}^{(i)}, h^-\rangle)}\right] \tag{5}$$

### 3.3 Fragment-based Predictive Pretraining

We further define two predictive pretraining tasks of which labels are extracted from the properties and the topologies of the fragment graphs. Only the molecular GNN is pretrained in this case. The labels provided by the fragment graphs guide the graph-based pretraining of the molecular GNNs, encouraging the model to learn higher-order structural information.

**Fragment Existence Prediction** A multi-label prediction task that outputs a vocabulary-size binary vector indicating which fragments exist in the molecular graph. Thanks to the optimized fragmentation procedure [20] that we use, the output dimension is compact without extremely rare classes or fragments, resulting in more robust learning.

**Fragment Graph Structure Prediction** We predict the structural backbones of fragment graphs. The number of classes is the number of unique structural backbones. Essentially, a backbone is a fragment graph with no node or edge attributes. Graphs share the same backbone if their fragments are arranged similarly. For example, fragment graphs in which three fragments connect in a line correspond to the same structural backbone, a line graph with three nodes. This task also benefits from the optimized fragmentation. As the extracted fragments are sufficiently large, the fragment graphs are small enough that the enumeration of all unique backbones is feasible.

The predictive tasks pretrain the graph embedding $h_G$. In particular, one task injects local community information of fragments while the other task injects the arrangements of fragments into $h_G$. We train both tasks together, optimizing the objective $\mathcal{L}_P = \mathcal{L}_{P_1} + \mathcal{L}_{P_1}$, where $\mathcal{L}_{P_1}$ and $\mathcal{L}_{P_2}$ are the predictive objective of each task. The predictive tasks are illustrated in Figure 2.

We can combine the predictive pretraining with the contrastive pretraining presented in Section 3.2. We found parallel pretraining more effective than the sequential pretraining described in [15]. With $\alpha$ being the weight hyperparameter, the joint pretraining objective is then:

$$\mathcal{L} = \alpha\mathcal{L}_P + (1-\alpha)\mathcal{L}_C \tag{6}$$

### 3.4 Combining Models based on Fragment Graphs and Molecule Graphs

We further propose to utilize both the molecule encoder $\text{GNN}_M$ and the fragment encoder $\text{GNN}_F$ for downstream prediction. Following the procedure described in Section 3.1, we can easily fragment and extract molecular graph and fragment graph representations from any unseen molecule. Such convenience is not always the case. For example, existing works that contrast 2D and 3D molecular graphs [23, 31] can only finetune the 2D encoder since 3D information is expensive to obtain for unseen molecules. Other works that rely on graph augmentation [35, 45, 46] cannot utilize the

augmented views as signals for downstream prediction since they are not faithful representations of the original molecule. In contrast, the representations extracted by our framework are faithful views that do not require expensive information other than the vanilla molecular structure.

Let $h_M$ and $h_F$ be the graph embeddings produced by $GNN_M$ and $GNN_F$, respectively. We obtain the downstream prediction by applying a fully-connected layer on the concatenation of $h_M$ and $h_F$.

## 4 Experiments

### 4.1 Experimental Settings

We pretrain GNNs according to the process discussed in section 3. The pretrained models are evaluated on various chemical benchmarks.

**Datasets**    We use a processed subset containing 456K molecules from the ChEMBL database [24] for pretraining. A fragment vocabulary of size 800 is extracted as described in Section 3.1.1. To ensure possible fragmentation of unseen molecules, we further complete the vocabulary with atoms not existing in the pretraining set, totaling 908 unique fragments. For downstream evaluation, we consider 8 binary graph classification tasks from MoleculeNet [36] with scaffold split [15]. Moreover, to assess the ability of the models in recognizing global arrangement, we consider two graph prediction tasks on large peptide molecules from the Long-range Graph Benchmark [7]. Long-range graph benchmarks are split using stratified random split. More information regarding datasets is provided in Appendix B.

**Models**    For graph classification benchmarks, we model our molecular encoders $GNN_M$ with the 5-layer Graph Isomorphism Network (GIN) [41] as in previous works [15, 23], using the same featurization. Similarly, we model the fragment encoder $GNN_F$ with a shallower 2-layer GIN since fragment graphs are notably smaller. For long-range benchmarks, both $GNN_M$ and $GNN_F$ are GIN with 5 layers, with the featurization of molecular graphs based on the Open Graph Benchmark [14]. All encoders have hidden dimensions of size 300. For more details, please refer to Appendix A.

**Pretraining and Finetuning**    All pretrainings are done in 100 epochs, with AdamW optimizer, batch size 256, and initial learning rate $1 \times 10^{-3}$. We reduce the learning rate by a factor of 0.1 every 5 epochs without improvement. We use the models at the last pretraining epoch for finetuning. On graph classification benchmarks, to ensure comparability, our finetuning setting is mostly similar to that of previous works [15, 23]: 100 epochs, Adam optimizer, batch size 256, initial learning rate $1 \times 10^{-3}$, and dropout rate chosen from $\{0.0, 0.5\}$. We reduce the learning rate by a factor of 0.3 every 30 epochs. On long-range benchmarks, the setting is similar except that we finetune for 200 epochs and factor the learning rate by 0.5 every 20 epochs without improvement. We find that using averaging as FRAGPOOL($\cdot$) works well. All experiments are run on individual Tesla $V100$ GPUs.

**Baselines**    We compare to several notable pretraining baselines, including predictive methods (AttrMask & ContextPred [15], G-Motif & G-Contextual (GROVER) [27]), generative method (GPT-GNN [16]), contrastive methods (GraphLoG [42], GraphCL [46], JOAO, JOAOvs [45]), contrastive method with privileged knowledge (GraphMVP [23]), and fragment-based method (MGSSL [49]). For long-range prediction, we compare our method with popular GNN architectures: GCN [19], GCNII [2], GIN [41], and GatedGCN [1] with and without Random Walk Spatial Encoding [6].

Our goal is to benchmark the quality of our proposed pretraining methods with existing work. The settings in our experiments are chosen to fulfill this objective. In general, our settings follow closely those from previous works to ensure comparability. Specifically, we use the same embedding model (5-layer GIN), featurization, and overall similar hyperparameters and amount of pretraining data as those in the baselines [15, 23, 42, 45, 46, 49]. We limit the amount of hyperparameter tuning for the same reason. In general, self-supervised learning and property prediction on molecular graphs are important research topics and a wide variety of methods have been proposed in terms of both the pretraining task and the learning model, resulting in impressive results [8, 50, 51]. Some of these performances stem from deeper neural network design, extensive featurization, and large-scale pretraining. Pretraining at the scale of [8, 51] requires hundreds of GBs or even a TB of memory.

Table 1: Test ROC-AUC on binary molecular property prediction benchmarks using different pre-training strategies in GraphFP. The top-3 performances on each dataset are shown in red color, with **red** being the best result, red being the second best result, and red being the third best result. The 2 rightmost column shows the average performance ranking (lower value means better ranking) and the average AUC. The last 5 rows show the performances of our methods, with $C$, $P$, and $F$ indicate contrastive pretraining, predictive pretraining, and inclusion of fragment encoders in downstream prediction, respectively.

| Pretraining Strategies | BBBP | Tox21 | ToxCast | SIDER | ClinTox | MUV | HIV | BACE | Avg. Rank | Avg. AUC |
|---|---|---|---|---|---|---|---|---|---|---|
| AttrMasking [15] | 64.3 ± 2.8 | **76.7 ± 0.4** | 64.2 ± 0.5 | 61.0 ± 0.7 | 71.8 ± 4.1 | 74.7 ± 1.4 | 77.2 ± 1.1 | 79.3 ± 1.6 | 7.88 | 71.15 |
| ContextPred [15] | 68.0 ± 2.0 | 75.7 ± 0.7 | 63.9 ± 0.6 | 60.9 ± 0.6 | 65.9 ± 3.8 | 75.8 ± 1.7 | 77.3 ± 1.0 | 79.6 ± 1.2 | 7.56 | 70.89 |
| G-Motif [27] | 66.9 ± 3.1 | 73.6 ± 0.7 | 62.3 ± 0.6 | 61.0 ± 1.5 | 77.7 ± 2.7 | 73.0 ± 1.8 | 73.8 ± 1.2 | 73.0 ± 3.3 | 14.25 | 70.16 |
| G-Contextual [27] | 69.9 ± 2.1 | 75.0 ± 0.6 | 62.8 ± 0.7 | 58.7 ± 1.0 | 60.6 ± 5.2 | 72.1 ± 0.7 | 76.3 ± 1.5 | 79.3 ± 1.1 | 11.88 | 69.34 |
| GPT-GNN [16] | 64.5 ± 1.4 | 74.9 ± 0.3 | 62.5 ± 0.4 | 58.1 ± 0.3 | 58.3 ± 5.2 | 75.9 ± 2.3 | 65.2 ± 2.1 | 77.9 ± 3.2 | 13.63 | 67.16 |
| GraphLoG [42] | 67.8 ± 1.9 | 75.1 ± 1.0 | 62.4 ± 0.2 | 59.5 ± 1.5 | 65.3 ± 3.2 | 73.6 ± 1.2 | 73.7 ± 0.9 | 80.2 ± 3.5 | 12.56 | 69.70 |
| GraphCL [46] | 69.7 ± 0.7 | 73.9 ± 0.7 | 62.4 ± 0.6 | 60.5 ± 0.9 | 76.0 ± 2.7 | 69.8 ± 2.7 | 78.5 ± 1.2 | 75.4 ± 1.4 | 12.13 | 70.78 |
| JOAO [45] | 70.2 ± 1.0 | 75.0 ± 0.3 | 62.9 ± 0.5 | 60.0 ± 0.8 | 81.3 ± 2.5 | 71.7 ± 1.4 | 76.7 ± 1.2 | 77.3 ± 0.5 | 9.56 | 71.89 |
| JOAOv2 [45] | 71.4 ± 0.9 | 74.3 ± 0.6 | 63.2 ± 0.5 | 60.5 ± 0.7 | 81.0 ± 1.6 | 73.7 ± 1.0 | 77.5 ± 1.2 | 75.5 ± 1.3 | 8.94 | 72.14 |
| GraphMVP [23] | 68.5 ± 0.2 | 74.5 ± 0.4 | 62.7 ± 0.1 | 62.3 ± 1.6 | 79.0 ± 2.5 | 75.0 ± 1.4 | 74.8 ± 1.4 | 76.8 ± 1.1 | 10.00 | 71.70 |
| MGSSL [49] | 68.9 ± 2.5 | 74.9 ± 0.6 | 63.3 ± 0.5 | 57.7 ± 0.7 | 67.5 ± 5.5 | 73.2 ± 1.9 | 75.7 ± 1.3 | 82.1 ± 2.7 | 10.94 | 70.41 |
| GraphFP-JT$_C$ | 71.5 ± 0.9 | 75.2 ± 0.5 | 63.6 ± 0.5 | 62.0 ± 1.0 | 77.7 ± 4.5 | 76.0 ± 2.2 | 75.6 ± 1.0 | 79.7 ± 1.3 | 12.66 | 72.66 |
| GraphFP-JT$_{CF}$ | 70.2 ± 1.7 | 72.7 ± 0.8 | 62.5 ± 0.9 | 59.3 ± 1.3 | 75.9 ± 5.6 | 73.9 ± 1.3 | 73.0 ± 1.9 | 74.2 ± 2.8 | 13.56 | 70.21 |
| GraphFP$_C$ | 71.5 ± 1.6 | 75.5 ± 0.4 | 63.8 ± 0.6 | 61.4 ± 0.9 | 78.6 ± 2.7 | 77.2 ± 1.5 | 76.3 ± 1.0 | 78.2 ± 3.4 | 5.50 | 72.81 |
| GraphFP$_P$ | 68.2 ± 1.2 | 76.0 ± 0.5 | 63.2 ± 0.7 | 59.3 ± 1.0 | 53.8 ± 3.8 | 74.5 ± 2.1 | 76.7 ± 1.0 | 80.7 ± 4.8 | 9.50 | 69.05 |
| GraphFP$_{CP}$ | 71.3 ± 1.7 | 75.5 ± 0.5 | 64.7 ± 0.2 | 61.3 ± 0.6 | 73.7 ± 3.9 | 76.6 ± 1.8 | 81.3 ± 2.2 | 81.3 ± 2.2 | 5.19 | 72.59 |
| GraphFP$_{CF}$ | 70.1 ± 1.8 | 74.3 ± 0.3 | 65.3 ± 0.8 | 64.7 ± 1.0 | 87.7 ± 5.8 | 74.5 ± 1.8 | 76.1 ± 2.0 | 77.1 ± 2.1 | 7.25 | 73.73 |
| GraphFP$_{CPF}$ | 72.0 ± 1.7 | 74.0 ± 0.7 | 63.9 ± 0.9 | 63.6 ± 1.2 | 84.7 ± 5.8 | 75.4 ± 1.9 | 78.0 ± 1.5 | 80.5 ± 1.8 | 4.56 | 74.01 |

Due to limited resources, we leave the evaluation of GraphFP and other baselines when pretrained on such larger datasets for future work.

## 4.2 Results on Graph Classification Benchmarks

Table 1 reports our results on chemical graph classification benchmarks. For each dataset, we report the mean and error from 10 independent runs with predefined seeds. Except for GraphLoG [42] and MGSSL [49], the results of other baselines are collected from the literature [15, 23, 37, 45, 46]. To ensure a comprehensive evaluation, we conduct experiments with all possible combinations of the proposed strategies, which include contrastive pretraining (denoted as $C$), predictive pretraining (denoted as $P$), and inclusion of fragment encoders in downstream prediction (denoted as $F$). Because $F$ requires $C$, all possible combinations of these components are $\{C, P, CP, CF, CPF\}$, with the corresponding pretrained models GraphFP$_C$, GraphFP$_P$, GraphFP$_{CP}$, GraphFP$_{CF}$, and GraphFP$_{CPF}$. For GraphFP$_{CP}$, we choose $\alpha = 0.3$ and for GraphFP$_{CPF}$, we choose $\alpha = 0.1$. The choices of $\alpha$ are reported in Appendix A. We also compare to GraphFP-JT$_C$ and GraphFP-JT$_{CF}$, which are variations of our models pretrained with the fragmentation from [18]. The fragments used by [18] are generally smaller, resulting in larger fragment graphs. We found 5-layer GIN models encode these larger graphs better. Our models are competitive in all benchmarks, obtaining the best performance in 5 out of 8 downstream datasets. We report the average ranking and the average AUC of the models. As reported in Table 1, the GNNs with our proposed fragment-based pretraining and finetuning strategies achieve the best average rankings and average AUCs across baselines. Moreover, adding the strategies successively, i.e. $C$, $CP$, and $CPF$, improves the average downstream rankings, confirming the individual effectiveness of each strategy in supporting the learning on molecular graphs. We attribute the remarkable performances of our methods to the capability of the pretrained embeddings and the fragment graph encoder in capturing higher-order structural patterns. For instance, the BBBP benchmark requires predicting blood-brain barrier permeability, where the molecular shape, size, and interaction with the transporting proteins play significant roles in determining the outcome. Capturing such information necessitates a deep understanding of the global molecular structure, which is the goal of our pretraining strategies.

## 4.3 Results on Long-range Chemical Benchmarks

In Table 2, we compare fragment-based pretraining and finetuning of GraphFP with GNN baselines on two long-range benchmarks: PEPTIDE-FUNC containing 10 classification tasks regarding peptide functions and PEPTIDE-STRUCT containing five regression tasks regarding 3D structural information [7]. Because fragment graphs of peptides are extremely large for effective extraction of structural backbones, we exclude predictive pretraining in this experiment. On both benchmarks, the results

Table 2: Performances on PEPTIDE-FUNC (graph classification) and PEPTIDE-STRUCT (graph regression). These tasks require capturing long-range interactions within large peptide molecules. Best performances are colored red.

| Methods | Peptide-func Test AP | Peptide-struct Test MAE |
|---|---|---|
| GCN | $0.5930 \pm 0.0023$ | $0.3496 \pm 0.0013$ |
| GCNII | $0.5543 \pm 0.0078$ | $0.3471 \pm 0.0010$ |
| GIN | $0.5498 \pm 0.0079$ | $0.3547 \pm 0.0045$ |
| GatedGCN | $0.5864 \pm 0.0077$ | $0.3420 \pm 0.0013$ |
| GatedGCN+RWSE | $0.6069 \pm 0.0035$ | $0.3357 \pm 0.0006$ |
| GraphFP$_{CF}$ | $0.6267 \pm 0.0073$ | $0.3137 \pm 0.0019$ |

Table 3: Effects on varying the size of the vocabulary

| Models | Fragmentation | | ROC-AUC | | |
|---|---|---|---|---|---|
| | Avg Frag Size | Avg Frag Graph Size | SIDER | ClinTox | HIV |
| GraphFP$_{800}$ | 2.40 | 4.61 | $61.4 \pm 0.9$ | $78.6 \pm 2.7$ | $76.3 \pm 1.0$ |
| GraphFP$_{1600}$ | 2.47 | 4.15 | $60.4 \pm 0.5$ | $76.4 \pm 4.6$ | $75.3 \pm 1.6$ |
| GraphFP$_{3200}$ | 2.53 | 3.73 | $60.6 \pm 0.5$ | $75.5 \pm 5.9$ | $75.1 \pm 1.3$ |

show that the proposed method outperforms other GNN baselines, including GatedGCN+RWSE with positional encoding. In particular, our pretrained model GraphFP$_{CF}$ obtains about 14% improvement on PEPTIDE-FUNC and 11.5% improvement on PEPTIDE-STRUCT compared to vanilla GIN. As the tasks require recognizing long-range structural information, these improvements indicate the effectiveness of our strategies in capturing the global arrangement of molecular graph components.

## 4.4 Analysis

**Pretraining with Various GNN Architectures**  In Table 4, we report the performances of some other GNN architectures besides GIN. The pretraining and finetuning conditions are similar to those described in Section 4.1 and Table 6. All architectures share the same input features, hidden dimensions, and the number of layers. On average, GIN, being the most expressive GNN among the ones shown, performs the best. This observation agrees with the conclusions from previous works [15, 41].

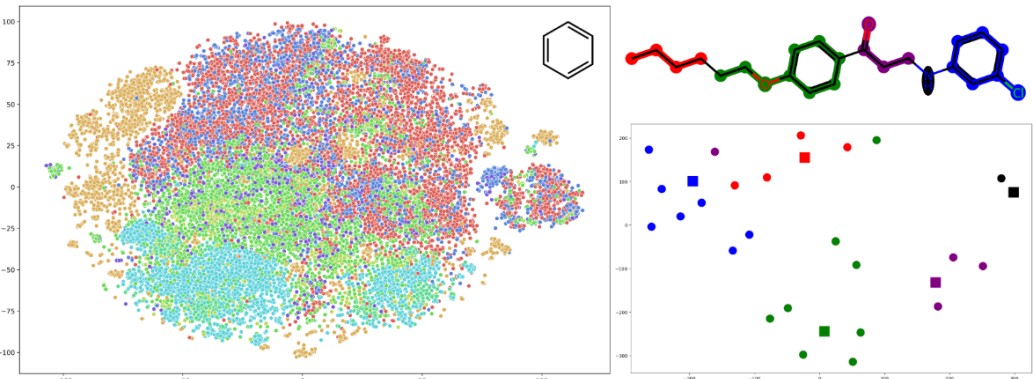

Figure 3: (left) t-SNE plot showing the embeddings of a common fragment. Each color corresponds to a structural backbone. (right) t-SNE plot showing the embeddings of atoms in a molecule with colors distinguishing fragment membership. Round markers indicate atom embeddings while square markers indicate fragment-based pooling of atom embeddings.

Table 4: Downstream performances with models pretrained with various GNN architectures.

| Architectures | BBBP | Tox21 | ToxCast | SIDER | ClinTox | MUV | HIV | BACE | Avg. Rank |
|---|---|---|---|---|---|---|---|---|---|
| GIN | $71.3 \pm 0.8$ | $75.8 \pm 0.3$ | $63.9 \pm 0.4$ | $61.3 \pm 0.4$ | $63.2 \pm 1.0$ | $76.2 \pm 2.0$ | $76.1 \pm 1.4$ | $82.5 \pm 1.9$ | 1.88 |
| GCN | $70.5 \pm 1.2$ | $74.3 \pm 0.3$ | $63.3 \pm 0.5$ | $61.5 \pm 0.8$ | $78.1 \pm 5.8$ | $77.8 \pm 1.8$ | $74.6 \pm 0.9$ | $76.2 \pm 2.1$ | 2.13 |
| GraphSage | $72.3 \pm 1.5$ | $74.5 \pm 0.6$ | $62.8 \pm 0.3$ | $61.2 \pm 0.8$ | $69.6 \pm 3.1$ | $77.3 \pm 1.3$ | $77.1 \pm 1.1$ | $78.0 \pm 3.6$ | 2.00 |

**Effects on Varying the Size of the Vocabulary**   The size of the fragment vocabulary likely influences the quality of pretraining and finetuning since it dictates the resolution of fragment graphs. To investigate such effects, we prepare two additional vocabularies with size $1,600$ and size $3,200$. We repeat the same contrastive pretraining as in Section 4.1 using the new vocabularies. As the vocabulary size increases, more unique and larger fragments are discovered, increasing the average fragment size and reducing the average fragment graph size. Table 10 shows that the performances worsen on some downstream benchmarks as the vocabulary size grows larger. We conjecture a few possible reasons. Firstly, a larger vocabulary means more parameters to optimize. Second, smaller fragment graphs represent an excessively loose view of the graph, resulting in a loss of structural information. In general, vocabulary size is an important hyperparameter that greatly affects the quality of self-supervised learning. In [20], the authors gave a discussion on selecting an optimal vocabulary size according to the entropy-sparsity trade-off.

**Visualizing the Learned Embeddings**   As shown in Figure 3, we visualize the learned embeddings via t-SNE [33]. The left plot illustrates the embeddings produced by the pretrained fragment encoder $GNN_F$ of a common fragment. Each dot corresponds to a molecule in which the fragment appears. The color of a dot indicates the structural backbone of the molecule it represents. For visibility, we only show the most common backbones. The embeddings are reasonably separated according to the structural backbones, indicating that the fragment embeddings capture higher-order structural information. Notice that the fragment encoder $GNN_F$ is not directly trained to recognize structural backbones. The right plot shows the embeddings produced by the molecule encoder $GNN_M$ of atoms within a molecule. The embeddings of atoms within the same fragment are clustered together. Interestingly, some clusters, shown in green, purple, and black, are arranged similarly to how they appear in the original molecule. This observation confirms that the collective embedding of nodes can capture higher-order connectivity.

# 5   Conclusions and Future Work

In this paper, we proposed contrastive and predictive learning strategies for pretraining GNNs based on graph fragmentation. Using an optimized fragment vocabulary, we pretrain two separate encoders for molecular graphs and fragment graphs, thus capturing structural information at different resolutions. When benchmarked on chemical and long-range peptide datasets, our method achieves competitive or better results compared to existing methods. Moving forward, we plan to further improve the pretraining via larger datasets, more extensive featurizations, better fragmentations, and more optimal representations. We also plan to extend the fragment-based techniques to other learning problems, including novel molecule generation and interpretability.

# Acknowledgments and Disclosure of Funding

This project is supported by the BioPACIFIC Materials Innovation Platform of the National Science Foundation under Award # DMR-1933487 and high-performance computing infrastructure from the California NanoSystem Institute at the University of California Santa Barbara.

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

# A   Experimental Settings

## A.1   Graph Isomorphism Network

Given a graph $G = (V, E)$ with node attributes $x_v$ for $v \in V$ and edge attributes $e_{uv}$ for $(u, v) \in E$, after the $k$-th layer of the GNN, the embedding of node $v$ is updated as:

$$h_v^{(k)} = U_k(h_v^{(k-1)}, M_k(\{(h_v^{(k-1)}, h_u^{(k-1)}, e_{uv}) | u \in N(v)\})), \tag{7}$$

where $N(v)$ is the set of neighbors of $v$ and $h_v^{(0)} = x_v$. The graph embedding $h_G$ is obtained via:

$$h_G = R(\{h_v^{(K)} | v \in V\}), \tag{8}$$

where $K$ is the number of layers and $R$ is the readout function. The overall graph embedding $h_G$ is used for graph-level prediction, as in Section 4.1. In our work as well as some previous works, Graph Isomorphism Network (GIN) with edge encoding [15, 23, 42] is often used as the underlying architectures of learning models. In particular:

$$h_v^{(k)} = \text{ReLU}\left(\text{MLP}^{(k)}\left(\sum_{u \in N(v) \cup \{v\}} h_v^{(k-1)} + \sum_{e \in \{(u,v) | v \in N(v) \cup \{v\}\}} h_e^{(k-1)}\right)\right), \tag{9}$$

where $h_e^{(k)}$ is the learnable edge embedding at the $k$-th layer. The mean readout is used to obtain the graph embedding:

$$h_G = \text{MEAN}(\{h_v^{(K)} | v \in V\}) \tag{10}$$

## A.2   Model and Training Configurations

The configurations of $\text{GNN}_M$ and $\text{GNN}_F$ are provided in Table 5. The pretraining and finetuning setups are provided in Table 6. In general, our parameters are comparable to those of previous works with which we compare [15, 23, 42]. Note that the number of node features and edge features is much fewer than that in [8, 50, 51]. On downstream tasks, we only finetune the dropout rate, selected from $\{0.0, 0.5\}$.

Table 5: Model Configuration

| Methods | Parameters | Values |
|---|---|---|
| $\text{GNN}_M$ | Convolution type | GIN |
| | Number of atom features | 2 |
| | Number of atom features for long-range benchmarks | 9 |
| | Number of edge features | 2 |
| | Number of edge features for long-range benchmarks | 3 |
| | Dimension of hidden embeddings | 300 |
| | Aggregation | SUM |
| | Number of layers | 5 |
| | Readout | MEAN |
| $\text{GNN}_F$ | Convolution type | GIN |
| | Number of fragment features | 1 |
| | Dimension of hidden embeddings | 300 |
| | Aggregation | SUM |
| | Number of layers | 2 |
| | Number of layers for long-range benchmarks | 5 |
| | Readout | MEAN |

## A.3   Combination of Predictive and Contrastive Pretraining

We combine the contrastive pretraining and predictive pretraining proposed in Section 3 by optimizing the joint objective function:

$$\mathcal{L} = \alpha \mathcal{L}_P + (1 - \alpha)\mathcal{L}_C, \tag{11}$$

Table 6: Training Configuration

| Phases | Hyperparameters | Values |
|---|---|---|
| Pretraining | Number of epochs | 100 |
| | Batch size | 256 |
| | Optimizer | AdamW |
| | Weight decay | 0.01 |
| | Learning rate | 0.001 |
| | Learning rate decay | Reduce on plateau |
| | Learning rate decay factor | 0.1 |
| | Learning rate decay patience | 5 epochs |
| Finetuning, chemical | Number of epochs | 100 |
| | Batch size | 256 |
| | Dropout | {0.0,0.5} |
| | Optimizer | Adam |
| | Weight decay | 0.0 |
| | Learning rate | 0.001 |
| | Learning rate decay | Step decay |
| | Number of steps before decay | 30 |
| | Learning rate decay | 0.3 |
| Finetuning, long-range | Number of epochs | 100 |
| | Batch size | 128 |
| | Dropout | {0.0,0.5} |
| | Optimizer | Adam |
| | Weight decay | 0.0 |
| | Learning rate | 0.001 |
| | Learning rate decay | Reduce on plateau |
| | Learning rate decay factor | 0.5 |
| | Learning rate decay patience | 20 epochs |

where $\mathcal{L}_P$ is the predictive objective, $\mathcal{L}_C$ is the contrastive objective, and $\alpha$ is the weight hyperparameter. We select $\alpha$ from $\{0.1, 0.2, 0.3, 0.4, 0.5\}$, using finetuning results as the performance indicator. Table 7 shows the performances using models trained with different values of $\alpha$.

Table 7: Downstream performances with models pretrained using different values of the weight hyperparameter $\alpha$. Subscripts $C$, $P$, and $F$ indicate contrastive pretraining, predictive pretraining, and inclusion of fragment encoders in downstream prediction, respectively. Best results for either $CP$ or $CPF$ models are shown in red. The rightmost column shows the average performance ranking for either $CP$ or $CPF$ models (lower value means better ranking).

| Models | BBBP | Tox21 | ToxCast | SIDER | ClinTox | MUV | HIV | BACE | Avg. Rank |
|---|---|---|---|---|---|---|---|---|---|
| CP, $\alpha = 0.1$ | 71.3 ± 0.8 | 75.8 ± 0.3 | 63.9 ± 0.4 | 61.3 ± 0.4 | 63.2 ± 1.0 | 76.2 ± 2.0 | 76.1 ± 1.4 | 82.5 ± 1.9 | 2.88 |
| CP, $\alpha = 0.2$ | 70.8 ± 1.9 | 75.3 ± 0.3 | 64.1 ± 0.3 | 62.1 ± 0.5 | 64.7 ± 7.9 | 76.5 ± 1.2 | 76.0 ± 1.1 | 83.2 ± 1.4 | 2.63 |
| CP, $\alpha = 0.3$ | 71.3 ± 1.7 | 75.5 ± 0.5 | 64.7 ± 0.2 | 61.3 ± 0.6 | 73.7 ± 3.9 | 76.6 ± 1.8 | 76.3 ± 1.0 | 81.3 ± 2.2 | 2.13 |
| CP, $\alpha = 0.4$ | 70.1 ± 2.3 | 76.1 ± 0.4 | 64.0 ± 0.3 | 61.2 ± 0.6 | 60.1 ± 6.7 | 75.6 ± 2.0 | 75.7 ± 1.0 | 82.1 ± 2.2 | 3.81 |
| CP, $\alpha = 0.5$ | 70.6 ± 1.1 | 76.1 ± 0.3 | 64.6 ± 0.5 | 59.2 ± 0.9 | 65.7 ± 4.2 | 74.1 ± 2.4 | 75.6 ± 1.1 | 81.9 ± 1.3 | 3.56 |
| CPF, $\alpha = 0.1$ | 72.0 ± 1.7 | 74.0 ± 0.7 | 63.9 ± 0.9 | 63.6 ± 1.2 | 84.7 ± 8.8 | 75.4 ± 1.9 | 78.0 ± 1.5 | 80.5 ± 1.8 | 1.63 |
| CPF, $\alpha = 0.2$ | 69.4 ± 1.3 | 73.7 ± 0.8 | 64.5 ± 0.4 | 63.3 ± 0.9 | 76.9 ± 3.2 | 73.0 ± 4.2 | 77.6 ± 1.3 | 79.9 ± 1.0 | 4.06 |
| CPF, $\alpha = 0.3$ | 70.0 ± 1.6 | 73.9 ± 0.5 | 65.0 ± 0.5 | 62.6 ± 0.9 | 83.0 ± 5.6 | 74.2 ± 2.7 | 77.6 ± 1.2 | 78.2 ± 1.0 | 3.25 |
| CPF, $\alpha = 0.4$ | 71.7 ± 0.9 | 73.3 ± 0.8 | 64.6 ± 0.5 | 63.5 ± 0.8 | 83.0 ± 6.9 | 73.3 ± 3.0 | 77.7 ± 2.1 | 78.8 ± 2.8 | 3.06 |
| CPF, $\alpha = 0.5$ | 70.3 ± 0.9 | 74.2 ± 0.7 | 64.8 ± 0.5 | 62.7 ± 0.9 | 79.3 ± 3.1 | 73.4 ± 2.8 | 76.7 ± 1.3 | 80.1 ± 1.2 | 3.00 |

# B  Datasets

Table 8 and Table 9 show statistics regarding the benchmark datasets used to evaluate our models. In Table 8, we list the number of learning instances and the number of binary tasks in each chemical dataset. Out of 8 datasets, 6 are small-size and 2 are medium-size. Table 9 presents long-range

datasets and predictive tasks on peptides. The peptide datasets have similar learning instances but with different predictive tasks.

Table 8: Binary Chemical Benchmarks

| Datasets | Descriptions | Number of Graphs | Number of Tasks |
|---|---|---|---|
| BBBP | Blood-brain barrier permeability | 2039 | 1 |
| Tox21 | Toxicology on 12 biological targets | 7831 | 12 |
| ToxCast | Toxicology measurements via high-throughput screening | 8575 | 617 |
| SIDER | Adverse drug reactions of marketed medicines | 1427 | 27 |
| ClinTox | Drugs that failed clinical trials for toxicity reasons | 1478 | 2 |
| MUV | Validation of virual screening techniques | 93087 | 17 |
| HIV | Ability to inhibit HIV replication | 41127 | 1 |
| BACE | Binding results for inhibitors of human $\beta$-secretase 1 | 1513 | 1 |

Table 9: Long-range Peptide Benchmarks

| Datasets | Peptides-func | Peptides-struct |
|---|---|---|
| Number of Graphs | 15535 | 15535 |
| Number of Tasks | 1 | 5 |
| Number of Classes | 10 | N/A |
| Task types | Multi-label classification | Multi-label regression |
| Descriptions | Peptide functions: antibacterial, antiviral, etc | 3D properties: length, sphericity, etc |

To maintain comparability with prior works, we use different featurizations for molecules from the binary chemical benchmarks and the long-range benchmarks. For data from Table 8, we follow the featurization used by [15], in which there are 2 input atom features and 2 input bond features [1]. For data from Table 9, we use the standard featurization provided in Open Graph Benchmark [14], in which there are 9 input atom features and 3 input bond features [2]. The latter featurization scheme is more up-to-date and adopted by recent works on machine learning in the chemical space [8, 50, 51].

## C  Vocabulary

### C.1  Vocabulary Statistics

In Figure 4, we present statistics on fragments in the extracted vocabulary, molecular graphs, and fragment graphs. Subplot **c** shows the distribution of fragments based on their sizes. The smallest fragments consist of singleton atoms, while the largest one contains 21 atoms. The majority of fragments comprise 1 to 14 atoms. Although the extracted vocabulary is precise, its fragments are both large and highly prevalent, which is not the case for fragmentations used in previous works. In fact, the most prevalent fragments are size-independent. As shown in subplot **d**, common fragments are well distributed among all fragment sizes. This high prevalence throughout the vocabulary fosters pattern recognition, thereby contributing to the competitive results achieved by our models. The presence of large fragments also positively impacts the size of the fragment graphs. Subplot **a** and subplot **b** depict the size distributions of the fragment graphs and the molecular graphs, respectively. Relative to the molecular graphs, fragment graphs are more sparse and considerably smaller in size. This property, combined with the optimized fragments, enables them to capture the higher-order connectivity patterns of graph components effectively.

### C.2  Vocabulary Size and Downstream Performance

The vocabulary size may influence the performance of pretrained models on downstream tasks. In Table 10, we report this effect when varying the vocabulary size from 200 to 3, 200, doubling the size with each step. The results suggest that in general, for each task, there is an optimal vocabulary size. For instance, GraphFP$_C$ performs the best on ClinTox and HIV when pretrained with a vocabulary of size 800. The optimal vocabulary size for Tox21 is 400 while the optimal size for BBBP falls

---

[1] Hu et al, ICLR 2020, Appendix C
[2] https://github.com/snap-stanford/ogb/blob/master/ogb/utils/features.py

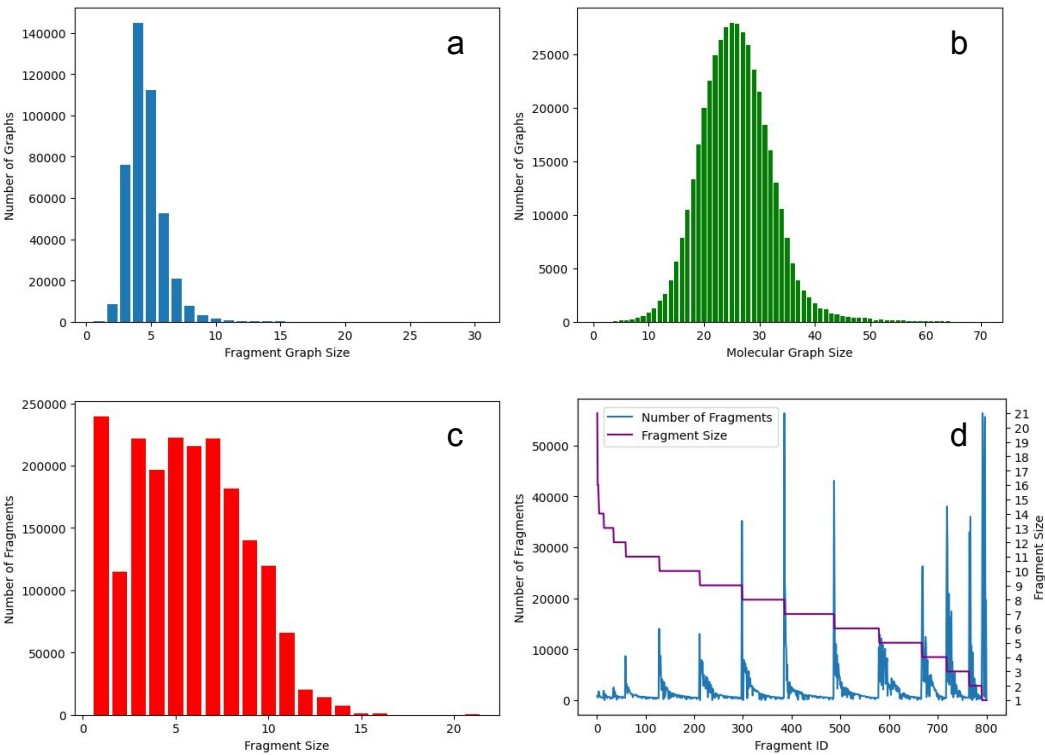

Figure 4: (a) Size distribution of fragment graphs (b) Size distribution of molecular graphs (c) Size distribution of fragments from the extracted vocabulary (d) Blue plot shows the occurrences of fragments in the vocabulary and purple plot shows the sizes of the fragments.

between $400$ to $800$. SIDER favors smaller vocabulary sizes while MUV favors larger vocabulary sizes. Varying vocabulary sizes seem to have little impact on the performance of ToxCast. BACE perceives the most interesting trend when the performance drops at vocabulary size $400$ and rises again when the size increases.

Table 10: Downstream performances with $GIN_C$ pretrained on vocabulary of various sizes.

| Vocab Size | BBBP | Tox21 | ToxCast | SIDER | ClinTox | MUV | HIV | BACE |
|---|---|---|---|---|---|---|---|---|
| 200 | $69.8 \pm 1.5$ | $75.6 \pm 0.8$ | $63.5 \pm 0.8$ | $61.3 \pm 0.7$ | $74.8 \pm 4.3$ | $74.9 \pm 1.8$ | $76.9 \pm 1.0$ | $79.4 \pm 1.6$ |
| 400 | $71.6 \pm 1.4$ | $75.8 \pm 0.5$ | $63.8 \pm 0.4$ | $61.3 \pm 0.7$ | $75.2 \pm 6.6$ | $75.7 \pm 2.5$ | $75.9 \pm 1.0$ | $77.5 \pm 2.5$ |
| 800 | $71.5 \pm 1.6$ | $75.5 \pm 0.4$ | $63.8 \pm 0.6$ | $61.4 \pm 0.9$ | $78.6 \pm 2.7$ | $77.2 \pm 1.5$ | $76.3 \pm 1.0$ | $78.2 \pm 3.4$ |
| 1600 | $71.1 \pm 1.6$ | $75.4 \pm 0.8$ | $63.9 \pm 0.9$ | $60.4 \pm 0.5$ | $76.4 \pm 4.6$ | $76.1 \pm 2.1$ | $75.3 \pm 1.6$ | $79.0 \pm 4.3$ |
| 3200 | $71.3 \pm 0.8$ | $75.4 \pm 0.4$ | $63.7 \pm 0.6$ | $60.6 \pm 0.5$ | $75.5 \pm 5.9$ | $77.1 \pm 1.9$ | $75.1 \pm 1.3$ | $79.4 \pm 4.5$ |

### C.3 Fragment Graph Statistics

We report several statistics regarding the fragment graphs from the prepared pretraining dataset described in Section 4.1. Table 11 lists the number of graphs for each fragment graph size. The smallest fragment graphs are of size 1 (standalone fragment) while the largest fragment graphs are of size 30. The majority of fragment graphs are of smaller sizes. This indicates that the fragmentation algorithm was able to extract large and frequent fragments, resulting in small fragment graphs that can capture higher-order connectivity.

To illustrate the connectivity within fragment graphs, for each fragment graph size, we report the average number of edges in Table 12. To save space, we only consider fragment graphs with a maximum size of 10. Since our molecular graphs are connected, our fragment graphs are also connected (i.e., no disconnected island). From the above table, we can see that, given a fragment graph with size $n$, the average number of edges connecting the fragments varies from around $n - 1$

Table 11: Number of Fragment Graphs by Size.

| Size | 1 | 2 | 3 | 4 | 5 | 6 | 7 | 8 | 9 | 10 | 11 | ... | 30 |
|---|---|---|---|---|---|---|---|---|---|---|---|---|---|
| Num Graphs | 167 | 8433 | 76024 | 144940 | 112426 | 52607 | 20999 | 7742 | 3334 | 1570 | 788 | ... | 30 |

to $n$. When the fragment graphs are small, the average number of edges is closer to $n-1$, indicating that the fragment graphs are mostly tree-like. As the fragment graph size increases, more loops appear and thus the average number of edges deviates further from $n-1$.

Table 12: Average number of edges by graph size.

| Size | 1 | 2 | 3 | 4 | 5 | 6 | 7 | 8 | 9 | 10 |
|---|---|---|---|---|---|---|---|---|---|---|
| Average Num Edges | 0.00 | 1.00 | 2.02 | 3.07 | 4.17 | 5.33 | 6.51 | 7.74 | 8.88 | 10.11 |

Finally, in Table 13, with varying vocabulary sizes, we report the average fragment graph size and the average number of edges for the whole pretraining dataset. As the vocabulary size increases, the size of fragment graphs decreases in general since larger vocabularies contain larger fragments.

Table 13: Average graph size and average number of edges with varying vocabulary size.

| Vocabulary Size | 200 | 400 | 800 | 1600 | 3200 |
|---|---|---|---|---|---|
| Average Graph Size | 5.92 | 5.17 | 4.61 | 4.15 | 3.73 |
| Average Num Edges | 5.23 | 4.40 | 3.78 | 3.28 | 2.84 |

