# OpenReview forum: "Fragment-based Pretraining and Finetuning on Molecular Graphs"
_NeurIPS.cc/2023/Conference — NeurIPS 2023 poster_

### Official Review · Reviewer_wcQm · 2023-06-25

**Soundness:** 3 good
**Presentation:** 3 good
**Contribution:** 3 good
**Rating:** 5
**Confidence:** 5

**Summary:**

In this paper, the authors proposed contrastively and predictively strategies for pretraining GNNs based on graph fragmentation. Using principle subgraph extraction, the authors pretrain two separate encoders for molecular and fragment graphs, capturing structural information at different resolutions.

**Strengths:**

1. The paper is well-organized and easy to follow.
2. The authors conduct comprehensive comparisons with baselines to show their advantages.
3. The fragment-level contrastive pretraining framework is novel, which captures both granular patterns and higher-order connectivity.
4. The t-SNE visualization in Figure 3 clearly shows the effectiveness of the authors' design.

**Weaknesses:**

1. The technical contribution is limited. For example, the principle subgraph extraction module is borrowed from [19].
2. The authors do not clearly state how to choose hyperparameter alpha.
3. In Table 3, the effects of the vocabulary size are only explored on three datasets.
4. The pertaining time is not reported in the paper.

**Questions:**

Please see the weakness.

**Limitations:**

The limitation is not clearly discussed. Please provide the discussion in the rebuttal period.

---

> ### Author Rebuttal · Authors · 2023-08-06
>
> Thank you for your time and insightful review. We are happy to answer your concerns.
>
> (1) **The technical contribution is limited. For example, the principle subgraph extraction module is borrowed from [19].**
>
> The focus of the paper is on developing pretraining and finetuning strategies based on the fragmen graphs. Our strategies are elegant and efficient, achieving improvement over recent strong baselines. The fragmentation component is not fixed and can be replaced with other similar frequency-based fragmentation methods. Please refer to point **(1)** from the main rebuttal for a detailed discussion regarding our contribution.
>
> (2) **The authors do not clearly state how to choose hyperparameter alpha.**
>
> We choose alpha from [0.1,0.5] with step size 0.1. The detailed performance on each value of alpha is reported in Table 6 from the Appendix.
>
> (3) **In Table 3, the effects of the vocabulary size are only explored on three datasets.**
>
> In **Table C** of the main rebuttal, we show the effect of varying vocabulary sizes on all 8 chemical benchmarks. We also added smaller vocabulary sizes (200, 400) as suggested by reviewer **RaR1**. We found interesting trends among datasets. Thank you for your suggestion.
>
> (4) **The pretaining time is not reported in the paper.**
>
> In general, it takes 4-5 hours to do one round of pretraining  (100 epochs) for any of our models on one V100 GPU. We did not find a significant difference in training time between the pretraining variations in our work. We will report the training time in the revision.
>
> (5) **The limitation is not clearly discussed.**
>
> This is a valid concern. In general, we do not observe any potential negative societal impact from our work. We do recognize a few limitations:
> - Our paper is empirical. The theoretical work for our paper is limited. We concluded the effectiveness of our designs based on downstream performances, ablation of training components, and visualization of the learned embeddings.
> - Due to resource constraints, we did not fully optimize the hyperparameters during pretraining and finetuning. We generally went with hyperparameters that seem to work well, referring to those used in previous work. Still, our models compare competitively with results from other works.
> - We pretrain on a subset of ChEMBL containing 456k molecules. Given that public database containing millions of molecules are available, the size of our pretraining data could be a bottleneck for performance. However, this pretraining size ($10^5$) is reasonable to compare with other baselines. GraphMVP used 50k molecules with 3D information for pretraining while others use pretraining set ranging from $10^5$ to $10^6$ in size.
> - We mainly evaluated on 8 popular binary prediction datasets. Molecular property prediction is a considerably rich field with a variety of datasets suitable for benchmarking. Even so, the benchmarks we used are the most popular among existing works.
> - We did not experiment with a variety of fragmentation methods. A recent paper [1], as pointed out by Reviewer **RaR1**, is another frequency-based fragmentation strategy comparable to the one used in our paper. It would be interesting to see how our methods perform with a variety of fragmentation methods, however, we'd like to leave this for future work.
>
> [1] Zijie Geng Z, Shufang Xie, Yingce Xia, et al. De Novo Molecular Generation via Connection-aware Motif Mining. ICLR 2023.

---

> > ### Comment · Reviewer_wcQm · 2023-08-15
> > **Reply to the authors**
> >
> > I have read the reply and appreciate the author's reply. My concerns are mostly resolved. Thanks!

---

> > > ### Author Response · Authors · 2023-08-16
> > > **Thank you. Any further questions?**
> > >
> > > Thank you for acknowledging our reply. We are glad that your concerns are mostly resolved. We are happy to discuss any further clarifications to improve your evaluation and enhance the acceptance chance of the paper.

---

### Official Review · Reviewer_Br6k · 2023-06-29

**Soundness:** 2 fair
**Presentation:** 3 good
**Contribution:** 2 fair
**Rating:** 5
**Confidence:** 3

**Summary:**

This paper presents a novel approach to pretrain Graph Neural Networks (GNNs) at the fragment level for property prediction on molecular graphs. By utilizing a compact vocabulary of prevalent fragments and introducing fragment-based contrastive and predictive pretraining tasks, the authors overcome the limitations of node-level and graph-level pretraining. Two different GNNs are pretrained to capture structural information at multiple resolutions, and the fragment information is utilized during finetuning. The proposed models show improved performances on both common molecular and long-range biological benchmarks.

**Strengths:**

- The paper is easy to follow.
- The idea that using motif for pretraining is novel and reasonable.



**Weaknesses:**

- Empirical performance is not strong enough. Especially in Table 2, it's hard to distinguish the absolute gain over the baselines. The authors are encouraged to report the average score over all tasks in molecular property prediction.
- The authors are also encouraged to compare with stronger baselines. For example, the authors can also compare JOAO V2 in addition to JOAO.
- Missing ablations: the authors add many components and loss functions in the system. It would be interesting know how each contribute to the performance.

**Questions:**

See weakness.

**Limitations:**

Yes

---

> ### Author Rebuttal · Authors · 2023-08-06
>
> Thank you for your time reviewing our paper and your thoughtful insights! We are happy to address your concerns.
>
> (1) **Empirical performance is not strong enough. The authors are encouraged to report the average score over all tasks in molecular property prediction.**
>
> We understand that the average score is often reported in the literature to give an evaluation of the overall performance across benchmarks. However, we argue that this metrics can be problematic since the values and variations of the scare can be quite different among benchmarks and a few benchmarks can skew the average value. Instead, we opt for rankings per benchmark and report the average ranking to evaluate overall performance.
>
> We do agree with the reviewer that absolute values give a better impression of the performance. As a result, we will add an extra column reporting this metric. Our 3 models, GIN_C, GIN_CP, and GIN_CPF, are the top 3 regarding both AUC and Ranking.
>
> | Baselines | AttrMasking | ContextPred | G-Motif | G-Contextual | GPT-GNN | GraphLoG | GraphCL |  JOAO | GraphMVP | MGSSL |   GIN_C   |   GIN_CP  |  GIN_CPF  |
> |:---------:|:-----------:|:-----------:|:-------:|:------------:|:-------:|:--------:|:-------:|:-----:|:--------:|:-----:|:---------:|:---------:|:---------:|
> | Avg. Rank |     5.81    |     5.44    |  10.06  |     8.69     |  10.38  |   8.94   |   8.69  |  7.31 |   6.00   |  8.13 |  **4.31** |  **3.69** |  **3.56** |
> |  Avg. AUC |    71.15    |    70.89    |  70.16  |     69.34    |  67.16  |   69.70  |  70.78  | 71.89 |   71.70  | 70.41 | **72.81** | **72.59** | **74.01** |
>
> In terms of the strength of the empirical evaluation, we argue that our models achieved substantial improvement over existing methods. Please see our response to the next concern.
>
> (2) **The authors are also encouraged to compare with stronger baselines. For example, the authors can also compare JOAO V2 in addition to JOAO.**
>
> The following table compares our models with JOAO and JOAOv2:
>
> |         | BBBP           | Tox21          | ToxCast        | SIDER          | ClinTox        | MUV            | HIV            | BACE           | Ave AUC |
> |---------|----------------|----------------|----------------|----------------|----------------|----------------|----------------|----------------|---------|
> | JOAO    | 70.2 ± 1.0     | 75.0 ± 0.3     | 62.9 ± 0.5     | 60.0 ± 0.8     | 81.3 ± 2.5     | 71.7 ± 1.4     | 76.7 ± 1.2     | 77.3 ± 0.5     | 71.89   |
> | JOAOv2  | 71.4 ± 0.9     | 74.3 ± 0.6     | 63.2 ± 0.5     | 60.5 ± 0.7     | 81.0 ± 1.6     | 73.7 ± 1.0     | 77.5 ± 1.2     | 75.5 ± 1.3     | 72.12   |
> | GIN_C   | 71.5 ± 1.6     | **75.5 ± 0.4** | 63.8 ± 0.6     | 61.4 ± 0.9     | 78.6 ± 2.7     | **77.2 ± 1.5** | 76.3 ± 1.0     | 78.2 ± 3.4     | 72.81   |
> | GIN_CPF | **72.0 ± 1.7** | 74.0 ± 0.7     | **63.9 ± 0.9** | **63.6 ± 1.2** | **84.7 ± 5.8** | 75.4 ± 1.9     | **78.0 ± 1.5** | **80.5 ± 1.8** | **74.01**   |
>
> In addition, Table B from the main rebuttal shows the average AUC across 8 chemical benchmarks of our models against strong baselines. The combination of our proposed pretraining strategies obtained 2.62%, 1.72%, and 1.29% relative improvement over JOAOv2, GraphMVP-G, and GraphMVP-C, respectively. When considering only the contrastive component, our model GIN-C (72.81) performs better than other contrastive models, including GraphCL (70.78), JOAOv2 (72.12), and GraphMVP (71.69).
>
> (3) **Missing ablations: the authors add many components and loss functions in the system. It would be interesting know how each contribute to the performance.**
>
> To clarify, we have conducted certain ablation on the components proposed in the paper. These components are:
> - C: contrastive pretraining.
> - P: predictive pretraining.
> - F: including fragment GNN in downstream prediction.
>
> Because F requires C, all possible combinations are {C, P, CP, CF, CPF}. In the paper, we included 3 out of 5 combinations (C, CP, CPF) and showed in Table 1 of the paper that more components successively improve the performance. We added results with all 5 combinations in Table A of the main rebuttal.

---

> > ### Comment · Reviewer_Br6k · 2023-08-15
> >
> > Dear Authors,
> >
> > Thank you for the detailed explanation and most of my concerns are addressed. I am happy to change my score. Also, please add these new numbers to your revision.
> >
> > Thanks,

---

> > > ### Author Response · Authors · 2023-08-15
> > > **Thank you**
> > >
> > > Thank you for your consideration. We appreciate your time reviewing and your help making the paper stronger. We will include the new numbers in the final revision.

---

### Official Review · Reviewer_MPQy · 2023-07-10

**Soundness:** 3 good
**Presentation:** 2 fair
**Contribution:** 2 fair
**Rating:** 6
**Confidence:** 3

**Summary:**

Based on the belief that learning with fragments can help capture structural information at multiple resolutions, this paper proposes a fragment-based strategy for pretraining and fine-tuning.

First, the paper extracts fragments by an existing heuristic algorithm called Principle Subgraph Mining algorithm to obtain fragments from a large molecular dataset (i.e.,  ChEMBL database). Then, the paper uses two separate GNNs (one for molecules and one for fragments) to learn the node embeddings. In particular, the node embeddings obtained by molecular-based GNN are pooled into fragment embeddings.
The model is trained with respect to three tasks: a contrastive task between fragment embeddings obtained by molecular-based GNN and  fragment-based GNN, a fragment existence prediction task, and a fragment graph structure prediction task.

The experiments are done on 8 binary graph classification tasks from MoleculeNet and 2 graph prediction tasks on large peptide molecules from the Long-range Graph Benchmark.

**Strengths:**

+ The proposed method is easy to follow and conduct.
+ The results on Long-range Graph Benchmark are particularly good.
+ Figure 1 clearly shows the method.
+ Codes are provided. Appendix further provides more details.

**Weaknesses:**

The idea of using molecular fragment can be interesting. The proposed method shows some effectiveness, although how it obtains can be less illuminating.

Many design choices need to be further described. Please reply to my questions below.

In addition, some writing issues exist. Sentences cannot start with "[reference]".

**Questions:**

1. The fragment extraction strategy can be described and compared with more details. The authors choose Principle Subgraph Mining algorithm to extract fragments. Do you try others? Any empirical evidence supports this choice? Like BRICKS or RECAP?
On line 53, the authors comment MGSSL as "their multi-step fragmentation may overly decomposes molecules, losing the ability to represent high-order structural patterns". Can you expand it? Why?
On line 122, the authors wrote that "however, existing methods use suboptimal fragmentation or fragment embeddings".  Likewise, can you provide more discussion about it? Why their methods obtain suboptimal fragments while yours can? Any theoretical proof?


2. The method assigns different fragment embeddings for two identical fragments appearing int the same molecule. Have you trying forcing them to be the same? I get that the local neighborhood of these two identical fragments can be different. But will it also help capture generic information if fragment is uniquely represented?


3. On lie 159, the authors wrote that "An edge exists between two fragment F nodes of GF is there exist at least a bond interconnecting atoms from the fragments." Will these cause too many edges between fragments? I suspect that most fragments are connected in this case. Therefore, the topology of fragment graph is lost.

4. How to obtain ground-truth structural backbones?

5. More experimental details of fragment extraction algorithm are needed. Any hyperparameters? On line 308, the authors wrote that "we prepare two additional vocabularies with size 1600 and size 3200". How do you prepare that?

6. Why results on long-range tasks can be much improved than existing works? Why different tasks use different baselines?

7. Figure 4(d) is hard to follow. How do you assign fragment ID? Why you need fragment ID as x axis?

**Limitations:**

No potential negative societal impact of their work as far as I know.

---

> ### Author Rebuttal · Authors · 2023-08-06
>
> Thanks for the thoughtful questions!
>
> (1) **The fragment extraction strategy can be described and compared with more details. ..., the authors comment MGSSL as "their multi-step fragmentation may overly decomposes molecules, losing the ability to represent high-order structural patterns"?**
>
> Rule-based methods (BRICS, RECAP) produce:
> - Very large vocabularies.
> - Most fragments have very low occurrence (like Kipf distribution).
> - Fragments with high occurrence are mostly small.
>
> A very large vocabulary with mostly rare and unique fragments is a major challenge for learning embeddings. One solution is further fragmenting the large and rare fragments to form a more concise vocabulary with mostly small fragments (MGSSL). However, if a fragment graph is constructed with small fragments, then this fragment graph is not much sparser than the original molecular graph. This means that the connectivity of the fragment graph is not much more global or higher-level than that of the molecular graph, which is the reason for our comment on MGSSL.
>
> We set the requirements for a fragment vocabulary to have precise size and contain larger fragments with good occurrences. Principal Subgraph Mining provides these qualities, so it is suitable for our study.
>
> (2) **On line 122, the authors wrote that "however, existing methods use suboptimal fragmentation or fragment embeddings". Likewise, can you provide more discussion about it?**
>
> We discuss these points from line 48 - 54. We'd like to further clarify them here:
> - The fragments in GROVER are k-hop subgraphs surrounding atom nodes, which limits the kind of patterns these fragments can represent because chemical patterns come in various sizes and shapes.
> - In MICRO-Graph, the fragment embeddings are contrasted with the graph embeddings. This encourages smoothing among the embeddings of fragments from the same graph since they all form positive pairs with the graph.
> - MGSSL, as stated above, overly decompose the fragments, hence their fragment graph cannot effectively represent high-order connectivity.
>
> We will add these explanations after line 122 in the revision.
>
> (3) **The method assigns different fragment embeddings for two identical fragments appearing in the same molecule. Have you tried forcing them to be the same?**
>
> Capturing structural information and multiple resolution and enforcing the consistency between node embeddings and fragment embeddings are our main objectives. The fragment embeddings distill onto the node embeddings 2 pieces of information: the local neighborhood, which comes from the fragment type, and the positional information with respect to the global arrangement, which comes from the embedding of higher-order connectivity via applying a GNN on the fragment graph. Both are arguably important for downstream prediction. Forcing the embeddings of 2 fragments of the same type to be similar will strengthen the first piece of information and weaken the second. Instead, we leaved the decision on weighting these 2 pieces of information to the contrastive learning and to the need of the downstream task. Generic information regarding each fragment is always captured because we keep an optimizable dictionary of fragment embeddings, which is tuned via learning.
>
> (4) **... I suspect that most fragments are connected in this case. Therefore, the topology of fragment graph is lost.**
>
> To clarify, there will be only 1 edge between a pair of connected fragments in the fragment graph no matter how many bonds there are connecting the fragments. To answer the reviewer's question: no, there will not be too many edges between fragments. Generally, fragment graphs are sparse and tree-like.
>
> (5) **How to obtain ground-truth structural backbones?**
>
> We record a dictionary of all possible structural backbones from the pretraining dataset with NetworkX's graph hashing algorithm. To obtain ground-truth structural backbone on an input graph, we remove all node features, hash the empty fragment graph, and look up the hash value in the dictionary.
>
> (6) **...the authors wrote that "we prepare two additional vocabularies with size 1600 and size 3200". How do you prepare that?**
>
> After each iteration of the fragmentation algorithm, a new larger fragment is added to the vocabulary by merging smaller ones, prioritizing frequency. The process repeats until the size of vocabulary reach a predefined value. To prepare larger vocabulary, we run the algorithm with more iterations.
>
> (7) **Why results on long-range tasks can be much improved than existing works? Why different tasks use different baselines?**
>
> Our method can effectively embed higher-order structural information. GNNs are inefficient in capturing long-range connectivity beyond a node's local neighborhood. In Table 2, GatedGCN with RWSE performs better than other GNNs because graph topological information (RWSE) is explicitly added to the node embeddings. Our pretraining effectively enforces this information into both node embeddings and fragment embeddings. To illustrate this point, Figure 3 shows that the fragment embeddings alone can well distinguish different structural backbones and node embeddings agree with the spatial arrangement within a molecule.
>
> For long-range tasks, we compared with GNNs to show that our pretrained GNN overcome the short-range problem of GNNs.
>
> (8) **Figure 4(d): How do you assign fragment ID? Why you need fragment ID as x axis?**
>
> The fragments are arranged based on decreasing size. The ID on the x-axis simply shows unique fragments in this order. The purple line shows the sizes of the fragments on the x-axis. Each horizontal plateau shows a collection of fragments with the same size. The blue plot is a histogram showing the frequency of each fragment with the spikes showing the most frequent fragments. These spikes are distributed across the x-axis, showing that most frequent fragments are of all sizes and not just small fragments.

---

> > ### Comment · Reviewer_MPQy · 2023-08-20
> > **Thanks for the reply. One question left.**
> >
> > Thanks for the reply.
> > Most of my concerns are cleared. But I DO have one question left.
> >
> > In my question (4), I know that there will be only 1 edge between a pair of connected fragments. Given your fragments are not very small, there can be many bonds interconnecting atoms from the fragments.  My point is, will this lead most of the fragments to be connected? Can you provide more detailed statistics, such as the average connected fragments?

---

> > > ### Author Response · Authors · 2023-08-20
> > > **Fragment graph statistics**
> > >
> > > We report several statistics regarding the fragment graphs.
> > >
> > > Number of fragment graphs by size:
> > > | Size |  1  |   2  |   3   |    4   |    5   |   6   |   7   |   8  |   9  |  10  |  11 |  12 | ... | 30 |
> > > |-|:-:|:-:|:-:|:-:|:-:|:-:|:-:|:-:|:-:|:-:|:-:|:-:|:-:|:-:|
> > > | Number of Graphs | 167 | 8433 | 76024 | 144940 | 112426 | 52607 | 20999 | 7742 | 3334 | 1570 | 788 | 496 | ... |  1 |
> > >
> > > The smallest fragment graphs are of size 1 (standalone fragment) while the largest fragment graphs are of size 30. The majority of fragment graphs are of smaller sizes.
> > >
> > > To illustrate the connectivity within fragment graphs, for each fragment graph size, we report the average number of edges. To save space, we only consider fragment graphs with maximum size 10:
> > >
> > > | Size |  1  |   2  |   3   |    4   |    5   |   6   |   7   |   8  |   9  |  10  |
> > > |-|:-:|:-:|:-:|:-:|:-:|:-:|:-:|:-:|:-:|:-:|
> > > | Average Number of Edges | 0.00 | 1.00 | 2.02 | 3.07 | 4.17 | 5.33 | 6.51 | 7.74 | 8.88 | 10.11 |
> > >
> > > Since our molecular graphs are connected, our fragment graphs are also connected (i.e, no disconnected island). From the above table, we can see that, given a fragment graph with size $n$, the average number of edges connecting the fragments vary from around $n-1$ to $n$. When the fragment graphs are small, the average number of edges are closer to $n-1$, indicating that the fragment graphs are mostly tree-like. As the fragment graph size increases, more loops appear and thus the average number of edges deviate further from $n-1$.
> > >
> > > To further illustrate the connectivity and sparsity within fragment graphs, in the following table, with varying vocabulary sizes, we show the average fragment graph size and the average number of edges:
> > >
> > > | Vocabulary Size |  200  |   400  |   800   |    1600   |    3200   |
> > > |-|:-:|:-:|:-:|:-:|:-:|
> > > | Average Graph Size | 5.92 | 5.17 |  4.61 | 4.15 |  3.73 |
> > > | Average Number of Edges | 5.23 | 4.40 | 3.78 | 3.28 | 2.84 |
> > >
> > > We hope that we could answer your question and improve your evaluation of the paper.
> > >
> > > We are happy to follow up if you have any other concern.

---

> > > > ### Comment · Reviewer_MPQy · 2023-08-21
> > > > **Thanks. I now raise my score.**
> > > >
> > > > Dear authors, thanks for the prompt reply. I have no further questions and raise my score to 6.

---

> > > > > ### Author Response · Authors · 2023-08-21
> > > > > **Thank you**
> > > > >
> > > > > Thank you for your helpful suggestions and re-evaluation!

---

### Official Review · Reviewer_Vdiv · 2023-07-11

**Soundness:** 3 good
**Presentation:** 3 good
**Contribution:** 3 good
**Rating:** 6
**Confidence:** 1

**Summary:**

The authors propose a novel method for generating representations for molecule graphs where two GNNs are contrastively learned. Using this new represntations, the authors achieve good results compared to a variety of baseline methods.

**Strengths:**

The paper and method are presented clearly.

The results are strong and the method is interesting + well explained

**Weaknesses:**

I found the presentation of Figure 3 a bit confusing

**Questions:**

i realise that many molecular benchmarks are based on similar organic compounds, but I am curious how the method would behave over a wider-range of molecules in material science.

Would it be possible to consider fragments are even larger common structures? Would this aid in longer range predictions?

> We reduce the learning rate by a factor of 0.1 every 5 epochs without improvement.
I'm curious if a different learning rate schedule would give better results. How is no improvement defined?

Long range structural information may involve 3D information, where. things that appear far apart on the graph may not be. I'm curious if this could be measured in some way? Could this method be applied to domains where 3D information is more directly used?

---

> ### Author Rebuttal · Authors · 2023-08-05
>
> Thank you for your time spent reviewing our paper! We appreciate the thoughtful comments and are happy to address your questions regarding the wider applicability of the model.
>
> (1) **I realise that many molecular benchmarks are based on similar organic compounds, but I am curious how the method would behave over a wider-range of molecules in material science.**
>
> This is an interesting question! Indeed learning and pretraining on material science compounds is an underexplored area compared to that of organic molecules. There are some differences between the two domains.
>
> There are a wide variety of materials and its hard to discuss all of them given the limitation of this forum. In general, we can say that while organic compounds are small and are stand-alone structures, materials have consituting units. Their properties are defined by both the arrangement of these units and the units themselves (for example: polymers with monomers, lattice with unit cells, and metal-organic frameworks). For application of GNNs to this domain, the graph representation needs to reflect these special characteristics. For example, [1] construct graphs for crystal lattices based on the connectivity within a unit cell. There have been investigations regarding the performance of GNNs on various types of material property prediction [2]. Most recently, several works have explored pretraining on materials, such as metal-organic frameworks [3] [4].
>
> Our method can be applied at different scale, to capture structural information either within the constituting units or of the unit arrangement. However, more research is needed to find an effective way of combining and utilizing these multiscale arrangement.  It would be interesting to see the performance of the method when applied to domains in which patterns are different than those in organic compounds but surely, some domain specific adjustments are needed. We are confident that it would be easier to adapt to materials in which organic patterns are presented such as polymers or metal-organic frameworks, but more investigation will need to be done for other types of compounds. In our work, we pretrained on 2 levels of topological resolution, but it is quite straightforward to extend to more levels.  We can have a separate loss to handle each pair of layer $n$ and layer $n+1$. However, as pretraining information can be distilled beyond 2 levels, we need a mechanism to control and tune this flow.
>
> (2) **Would it be possible to consider fragments are even larger common structures? Would this aid in longer range predictions?**
>
> It would be quite interesting to do so. However, a major problem is that as the size of substructures grows, they become rarer and more unique because in general, smaller substructures are more likely to repeat (think singular atoms, bonds, or simple rings). Even when we could extract high-frequency large fragments, this is not always useful, as we discussed in Section 4.4 and showed in Table 3. Larger fragments may overly "blur" the higher order connectivity, leading to worsen performance.
>
> There are cases when we indeed want much larger fragments, such as when working with macromolecules. In those cases, the substructure rarity problem still exists. One possible idea is to look for common substructures within the fragment graphs and graphs with even higher order of connectivity and enforce this information up and down the hierarchy. This expansion may lead to better performance on long-range prediction.
>
> (3) **I'm curious if a different learning rate schedule would give better results. How is no improvement defined?**
>
> No improvement means no new minima in the loss is found after the given number of epochs (5 in this case).
>
> We believe the performance will likely be better with learning rate tuning. However, due to limited resource, we generally went with hyperparameters that seems to work well. The hyperparameters we used are similar and comparable to those of the baselines. The factor value of 0.1 is typical in practice, and the patient value of 5 epochs came from our observation of a few initial training cycles.
>
> (4) **Long range structural information may involve 3D information, where. things that appear far apart on the graph may not be. I'm curious if this could be measured in some way? Could this method be applied to domains where 3D information is more directly used?**
>
> 3D coordinates are valuable information that would likely help property prediction. However, such information is expensive to obtain on a large scale. When it is available, pretraining can help distill such knowledge into the molecule's embeddings, like in GraphMVP [5] or 3D-InfoMax [6]. In that case, our learned embeddings, which encode long-range information, likely adapt well to incorporating 3D information.
>
> [1] Xie, T., & Grossman, J. C. (2018). Crystal graph convolutional neural networks for an accurate and interpretable prediction of material properties. Physical review letters, 120(14), 145301.
>
> [2] Fung, V., Zhang, J., Juarez, E., & Sumpter, B. G. (2021). Benchmarking graph neural networks for materials chemistry. npj Computational Materials, 7(1), 84.
>
> [3] Cao, Z., Magar, R., Wang, Y., & Barati Farimani, A. (2023). Moformer: self-supervised transformer model for metal–organic framework property prediction. Journal of the American Chemical Society, 145(5), 2958-2967.
>
> [4] Kang, Y., Park, H., Smit, B., & Kim, J. (2023). A multi-modal pre-training transformer for universal transfer learning in metal–organic frameworks. Nature Machine Intelligence, 5(3), 309-318.
>
> [5] Liu, S., Wang, H., Liu, W., Lasenby, J., Guo, H., & Tang, J. (2021). Pre-training molecular graph representation with 3d geometry. arXiv preprint arXiv:2110.07728.
>
> [6] Stärk, H., Beaini, D., Corso, G., Tossou, P., Dallago, C., Günnemann, S., & Liò, P. (2022, June). 3d infomax improves gnns for molecular property prediction. In International Conference on Machine Learning (pp. 20479-20502). PMLR.

---

### Official Review · Reviewer_RaR1 · 2023-07-27

**Soundness:** 3 good
**Presentation:** 3 good
**Contribution:** 2 fair
**Rating:** 6
**Confidence:** 4

**Summary:**

This paper proposes a contrastively and predictively strategy for pretraining GNNs based on graph fragmentation. Specifically, it leverages a frequency-based method for extracting molecule fragments, and performs fragment-based contrastive and predictive tasks to jointly pretrain two GNNs on a molecule graph and a fragment graph. It also enforces the consistency between fragment embeddings and atom embeddings for multi-solution structural information.

**Strengths:**

1.	The paper is easy to follow.
2.	The paper investigates an interesting fragmentation strategy for pretraining tasks.
3.	The proposed method enforces the consistency between fragment embeddings and atom embeddings for multi-solution structural information, which is a promising trick. Experiments demonstrate the effectiveness of this method.

**Weaknesses:**

1.	The technical novelty is limited, because it is a combination of existing methods. While the performance improvement is not very remarkable.
2.	The authors may want to conduct ablation studies on the effect of molecule fragmentation strategy and the pretraining strategy, respectively.
3.	Table 3 shows that the performances worsen on some downstream benchmarks as the vocabulary size grows larger. The authors may want to investigate a smaller vocabulary size. When it reaches 1, the method is the same as without fragments.
4.	A related work [1]---which leverages a similar frequency-based motif extractor and uses contrastive learning for generative training---is missing.

[1] Zijie Geng Z, Shufang Xie, Yingce Xia, et al. De Novo Molecular Generation via Connection-aware Motif Mining. ICLR 2023.

**Questions:**

1.	The results of MGSSL [1] is different from what they report in the original paper. Why?
2.	Do you kekulize the molecules? Or how do your deal with the possible breakage of aromatic systems?
3.	Why do you use a subset of ChEMBL for pretraining? How is the dataset processed?

[1] Zaixi Zhang, Qi Liu, Hao Wang, Chengqiang Lu, and Chee-Kong Lee. Motif-based graph self-supervised learning for molecular property prediction. NeurIPS 2021.

---

> ### Author Rebuttal · Authors · 2023-08-06
>
> Thank you for your thoughtful suggestions! We are happy to address your concerns.
>
> (1) **The technical novelty is limited, because it is a combination of existing methods**
>
> We would like to argue that our work is not simply a combination of existing methods. Besides Micro-Graph [1], ours is the only work that does fragment-level contrastive pretraining. Our proposed fragment-based pretraining and finetuning strategies are novel and efficient. We used an existing fragmentation procedure, however, the fragmentation component used is not fixed and can be replaced with other frequency-based fragmentation methods. Please refer to point **(1)** from the main rebuttal for a detailed discussion regarding our contributions.
>
> (2) **The performance improvement is not very remarkable.**
>
> For comparison, **Table B** from the main rebuttal the average AUC across 8 common chemical benchmarks of our models against strong baselines. The combination of our proposed pretraining strategies obtained 2.62%, 1.72%, and 1.29% relative improvement over JOAOv2, GraphMVP-G, and GraphMVP-C, respectively. When considering only the contrastive component, our model GIN-C (72.81) performs better than other contrastive models, including GraphCL (70.78), JOAOv2 (72.12), and GraphMVP (71.69).
>
> (3) **The authors may want to conduct ablation studies on the effect of molecule fragmentation strategy and the pretraining strategy.**
>
> To clarify, we have conducted certain ablation on the components proposed in the paper, which are:
> - C: contrastive pretraining.
> - P: predictive pretraining.
> - F: including fragment GNN in downstream prediction.
>
> Because F requires C, all possible combinations are {C, P, CP, CF, CPF}. In the paper, we included 3/5 combinations (C, CP, CPF) and showed that more components successively improve the performance. We provide the full results with 5 combinations (**Table A** in the main rebuttal). Given the limit of the rebuttal period and the focus of our paper on the pretraining and finetuning instead of the fragmentation, we leave the ablation on the fragmentation strategies for future work.
>
> (4) **The authors may want to investigate a smaller vocabulary size. When it reaches 1, the method is the same as without fragments.**
>
> To clarify, the vocabulary size cannot go to 1 since it should at least contain unique singular atom fragments. We added 2 additional experiments with smaller vocabulary sizes (200,400) and show the full results on 8 benchmarks (**Table C** in the main rebuttal). Thank you for your suggestion.
>
> (5) **A related work [1] ... is missing.**
>
> Thank you for the pointer. We will include this citation in the revision.
>
> (6) **The results of MGSSL [1] is different from ... the original paper. Why?**
>
> We rerun MGSSL and GraphLoG using the codes provided by the authors to ensure the results are comparable to other baselines (10 runs). The results from MGSSL paper used 3 runs and the results from GraphLoG are from the last epoch, not the best validation epoch.
>
> (7) **Do you kekulize the molecules? Or how do your deal with the possible breakage of aromatic systems? Why do you use a subset of ChEMBL for pretraining? How is the dataset processed?**
>
> The benchmark datasets and pretraining dataset are curated data from existing works. These datasets have been cleaned in a standard manner, including kekulization. The size of our pretraining dataset (456k) is comparable to those used to pretrain the baseline methods and is optimal for our limited computing resource. As far as we know, the motif mining algorithm does not produce fragments with incomplete aromatic system. Even so, our pipeline is robust to such cases because of how the fragment graphs are constructed. We have a single edge connecting two neighboring fragments. This edge is non-featurized and no matter the type of connection between two fragments (single bond, multiple bonds, overlapping atoms, etc), there is only one edge connecting them. Essentially, the fragment graphs only contain high-level arrangement within molecules and not granular bonding information. As a result, the pipeline is robust to cases in which fragments may contain incomplete aromatic system, such as fragmentation based on functional groups.
>
>
> [1] Zhang, S., Hu, Z., Subramonian, A., & Sun, Y. (2020). Motif-driven contrastive learning of graph representations. arXiv preprint arXiv:2012.12533.

---

> > ### Comment · Reviewer_RaR1 · 2023-08-15
> > **Thanks for the response. Increasing my score from 4 to 6.**
> >
> > I appreciate the authors' efforts in responding to my concerns. I have raised my score from 4 to 6.
> >
> > I understand the technical novelty of the paper now, and I admit that it has some interesting insights, especially the cross-level contrastive learning between node embeddings and fragment embeddings. Although I still think these techniques mainly involve detailed manners and are a bit incremental.
> >
> > I understand that a frequency-based motif extractor is suitable for the model, but a comparison with commonly seen motif extractors such as that in JT-VAE will make the paper stronger. Considering the time limitation during rebuttal, I expect the authors to include the comparison in a future revision.

---

> > > ### Author Response · Authors · 2023-08-15
> > > **Thank you**
> > >
> > > Thank you for your consideration. We are happy that we have addressed your concerns. Your suggestion is valuable to us and we will conduct further experiments with existing fragmentations, such as the one used in JT-VAE, in the final revision.

---

### Author Rebuttal · Authors · 2023-08-06

We want to thank the reviewers for their time and insightful comments!
We would like to use this space to summarize and address some common concerns. Citations follow those in the paper.

**(1)** Reviewer **RaR1** and Reviewer **wcQm** raise concerns about the level of contribution of our work.  Reviewer **wcQm** reasoned that the lack of novelty comes from the borrowed fragmentation technique from another work [19]. We used [19] because this algorithm is suitable for our requirements. The fragmentation component is not fixed and can be replaced with other frequency-based fragmentation methods. Our contribution is in the pretraining and finetuning strategies with fragment graphs. For that reason, we respectfully disagree with reviewer **RaR1**'s statement that the paper is a combination of existing methods. We studied fragment-level pretraining, an interesting direction where few prior works exist [26], [46], [47]. Ours is the only beside [46] that does fragment-level contrastive pretraining. Compared to previous work, we proposed to:
- Conduct cross-level contrastive learning between node embeddings and fragment embeddings.
- Learn separate sets of embeddings for node and fragments and train separate models for molecular graphs and fragment graphs.
- Contrast fragment embeddings with aggregated embeddings of nodes corresponding to the fragments. This design allows much more flexibility in learning multi-resolution topology compared to those of previous works.
- Utilize the fragment graphs in both predictive pretraining and downstream prediction.

Our methods are not only elegant, but also efficient. Compared to [47] which took 20 hours to train, each of our models took less than 5 hours to train in total (using similar hardware) while obtaining superior performances.

**(2)** Reviewer **RaR1** and Reviewer **Br6k** indicated a lack of ablation study regarding the various components we proposed. In particular, we have 3 unique components:
- C: contrastive pretraining using separate molecule GNN and fragment GNN.
- P: predictive pretraining utilizing information from fragment graphs.
- F: including fragment GNN in downstream prediction.

To clarify, we have conducted considerable ablation on these components. Because F requires C, all possible combinations of these components are {C, P, CP, CF, CPF}. In the paper, we included 3/5 combinations (Table 1: C, CP, CPF) and showed that more components successively improve the downstream performance, confirming the positive contribution of each component. We report results on all combination in Table A. We only include the average AUC and ranking (among the combinations) here and will include the full details in the final revision.

**Table A**

|Model|GIN_C|GIN_P|GIN_CP|GIN_CF|GIN_CPF|
|:-:|:-:|:-:|:-:|:-:|:-:|
|Avg AUC|72.81|69.05|72.59|73.73|74.01|
|Avg Rank|2.88|3.69|2.75|3.19|2.50|

C and F are stronger contributors than P, however, when combined with others, P has positive impacts on the overall performance. Disparity between AUC and Rank is because strong performance on a few benchmarks can skew the average AUC. Between GIN_CP and GIN_CF, GIN_CF has better AUC because it did particularly well on ClinTox and ToxCast but GIN_CP has better overall ranking because it has more pretraining than GIN_CF.

**(3)** Reviewer **RaR1** and Reviewer **Br6k** raised questions regarding the strength of our results and improvements. For comparison, the following Table B shows the average AUC across 8 common chemical benchmarks. GraphMVP-G and GraphMVP-C are recent strong baselines, and JOAO-v2 is a strong contrastive baseline suggested by Reviewer **Br6k**.

**Table B**
|Baselines|GraphCL|MGSSL|JOAOv2|GraphMVP|GraphMVP-G|GraphMVP-C|GIN_C|GIN_CPF|
|:-:|:-:|:-:|:-:|:-:|:-:|:-:|:-:|:-:|
|Avg AUC|70.78|70.41|72.12|71.69|72.76|73.07|72.81|74.01|

The combination of our proposed pretraining strategies in GIN_CPF obtained 2.62%, 1.72%, and 1.29% relative improvement over JOAOv2, GraphMVP-G, and GraphMVP-C, respectively. Notice that GraphMVP relies on 3D information, which is expensive to obtain and difficult to scale. Compared to generative methods (MGSSL and GraphMVP-G), our strategies are much faster to train. MGSSL takes 20 hours while GIN-CPF takes less than 5 hours to train. Considering only the contrastive component, our model GIN-C (72.81) performs better than others, including GraphCL (70.78), JOAOv2 (72.12), and GraphMVP (71.69).

**(4)** Reviewer **wcQm** and reviewer **RaR1** recommend expanding the analysis on the effects of the vocabulary size. This is a valuable suggestion and we are happy to address it. In particular, we added 2 smaller vocabulary with sizes 200 and 400 to the analysis and report the downstream performances of GIN_C on all 8 datasets in Table C.

**Table C**

| Vocab Size |BBBP|Tox21|ToxCast|SIDER|ClinTox|MUV|HIV|BACE|
|:-:|:-:|:-:|:-:|:-:|:-:|:-:|:-:|:-:|
|200|69.8±1.5|75.6±0.8|63.5±0.8|61.3±0.7|74.8±4.3|74.9±1.8|76.9±1.0|79.4±1.6|
|400|71.6±1.4|75.8±0.5|63.8±0.4|61.3±0.7|75.2±6.6|75.7±2.5|75.9±1.0|77.5±2.5|
|800|71.5±1.6|75.5±0.4|63.8±0.6|61.4±0.9|78.6±2.7|77.2±1.5|76.3±1.0|78.2±3.4|
|1600|71.1±1.6|75.4±0.8|63.9±0.9|60.4±0.5|76.4±4.6|76.1±2.1|75.3±1.6|79.0±4.3|
|3200|71.3±0.8|75.4±0.4|63.7±0.6|60.6±0.5|75.5±5.9|77.1±1.9|75.1±1.3|79.4±4.5|

The size of the vocabulary directly influences the size of fragments. Small fragments may result in fragment graphs that are too granular to capture high-level connectivity effectively. On the other hand, fragments being too large may lead to an overly loose view of the graph and a loss of structural information. The results suggest that in general, for each task, there is an optimal sweet spot for vocabulary size for which it should be finetuned. For example, the optimal vocabulary size for Tox21 is 400 while the optimal size for BBBP falls between 400 to 800. SIDER favors smaller vocabulary sizes while MUV favors larger vocabulary sizes.

---

### Decision · Program_Chairs · 2023-09-21

**Decision:**

Accept (poster)

**Comment:**

The authors proposed a contrastive learning algorithm for molecular representation learning. To split the molecular graph into fragments, the authors use a method similar to BPE in NLP. The constrastive learning is conducted between fragment graphs and molecule graphs.

Following thorough discussions among reviewers, the consensus leans towards acceptance. Consequently, I have decided to adhere to the collective decision and recommend acceptance.

However, I DO have the following concerns and I hope the authors should discuss / address in the camera ready version:

- The rationale of the constrastive learning between a molecular graph and fragment graph is not clear. What is the advantage over the constrastive learning within molecular graphs? Any experimental results?


- L246: 456K molecules and 5-layer GIN for pre-training: This is not convincing to me. Pre-training should be conducted on large amount of data and large models, while the results in this paper are not convincing.

- The results on MoleculeNet seem to be far behind SOTA.

  -  https://www.nature.com/articles/s42256-021-00438-4

   - https://arxiv.org/pdf/2207.08806.pdf or https://dl.acm.org/doi/10.1145/3534678.3539368

    -  I understand the above two works use both 2D and 3D information. However, given so many molecular pre-training papers, will the proposed method in this paper benefit and influence more readers?

   - Even if for the work not using 3D information, https://dl.acm.org/doi/pdf/10.1145/3580305.3599317, this works gives a much stronger baseline than that in the current submission.

  - I think the authors should discuss with the above works.